# On the Modularity of Hypernetworks

**Tomer Galanti**
School of Computer Science
Tel Aviv University
`tomerga2@tauex.tau.ac.il`

**Lior Wolf**
Facebook AI Research (FAIR) &
Tel Aviv University
`wolf@fb.com`

## Abstract

In the context of learning to map an input $I$ to a function $h_I : \mathcal{X} \to \mathbb{R}$, two alternative methods are compared: (i) an embedding-based method, which learns a fixed function in which $I$ is encoded as a conditioning signal $e(I)$ and the learned function takes the form $h_I(x) = q(x, e(I))$, and (ii) hypernetworks, in which the weights $\theta_I$ of the function $h_I(x) = g(x; \theta_I)$ are given by a hypernetwork $f$ as $\theta_I = f(I)$. In this paper, we define the property of modularity as the ability to effectively learn a different function for each input instance $I$. For this purpose, we adopt an expressivity perspective of this property and extend the theory of [6] and provide a lower bound on the complexity (number of trainable parameters) of neural networks as function approximators, by eliminating the requirements for the approximation method to be robust. Our results are then used to compare the complexities of $q$ and $g$, showing that under certain conditions and when letting the functions $e$ and $f$ be as large as we wish, $g$ can be smaller than $q$ by orders of magnitude. This sheds light on the modularity of hypernetworks in comparison with the embedding-based method. Besides, we show that for a structured target function, the overall number of trainable parameters in a hypernetwork is smaller by orders of magnitude than the number of trainable parameters of a standard neural network and an embedding method.

## 1 Introduction

Conditioning refers to the existence of multiple input signals. For example, in an autoregressive model, where the primary input is the current hidden state or the output of the previous time step, a conditioning signal can drive the process in the desired direction. When performing text to speech with WaveNets [35], the autoregressive signal is concatenated to the conditioning signal arising from the language features. Other forms of conditioning are less intuitive. For example, in Style GANs [16], conditioning takes place by changing the weights of the normalization layers according to the desired style.

In various settings, it is natural to treat the two inputs $x$ and $I$ of the target function $y(x, I)$ as nested, i.e., multiple inputs $x$ correspond to the 'context' of the same conditioning input $I$. A natural modeling [4, 30, 27] is to encode the latter by some embedding network $e$ and to concatenate it to $x$ when performing inference $q(x, e(I))$ with a primary network $q$. A less intuitive solution, commonly referred to as a hypernetwork, uses a primary network $g$ whose weights are not directly learned. Instead, $g$ has a fixed architecture, and a second network $f$ generates its weights based on the conditioning input as $\theta_I = f(I)$. The network $g$, with the weights $\theta_I$ can then be applied to any input $x$.

Hypernetworks hold state of the art results on numerous popular benchmarks [1, 36, 2, 37, 22], especially due to their ability to adapt $g$ for different inputs $I$. This allows the hypernetwork to model tasks effectively, even when using a low-capacity $g$. This lack of capacity is offset by using very large networks $f$. For instance, in [21], a deep residual hypernetwork that is trained from scratch

outperforms numerous embedding-based networks that rely on ResNets that were pre-trained on ImageNet.

The property of modularity means that through $f$, the network $g$ is efficiently parameterized. Consider the case in which we fit individual functions $g_I'$ to model each function $y_I = y(\cdot, I)$ independently (for any fixed $I$). To successfully fit any of these functions, $g_I'$ would require some degree of minimal complexity in the worst case. We say that modularity holds if the primary network $g$ whose weights are given by $f(I)$ has the same minimal complexity as required by $g_I'$ in the worst case.

In this paper, we seek to understand this phenomenon. For this purpose, we compare two alternatives: the standard embedding method and the hypernetwork. Since neural networks often have millions of weights while embedding vectors have a dimension that is seldom larger than a few thousand, it may seem that $f$ is much more complex than $e$. However, in hypernetworks, often the output of $f$ is simply a linear projection of a much lower dimensional bottleneck [21]. More importantly, it is often the case that the function $g$ can be small, and it is the adaptive nature (where $g$ changes according to $I$) that enables the entire hypernetwork ($f$ and $g$ together) to be expressive.

In general, the formulation of hypernetworks covers embedding-based methods. This implies that hypernetworks are at least as good as the embedding-based method and motivates the study of whether hypernetworks have a clear and measurable advantage. Complexity analysis provides a coherent framework to compare the two alternatives. In this paper, we compare the minimal parameter complexity needed to obtain a certain error in each of the two alternatives.

**Contributions**    The central contributions of this paper are: **(i)** Thm. 1 extends the theory of [6] and provides a lower bound on the number of trainable parameters of a neural network when approximating smooth functions. In contrast to previous work, our result does not require that the approximation method is robust. **(ii)** In Thms. 2-4, we compare the complexities of the primary functions under the two methods ($q$ and $g$) and show that for a large enough embedding function, the hypernetwork's primary $g$ can be smaller than $q$ by orders of magnitude. **(iii)** In Thm. 5, we show that under common assumptions on the function to be approximated, the overall number of trainable parameters in a hypernetwork is much smaller than the number of trainable parameters of a standard neural network. **(iv)** To validate the theoretical observations, we conducted experiments on synthetic data as well as on self-supervised learning tasks.

To summarize, since Thm. 1 shows the minimal complexity for approximating smooth target functions, and Thm. 4 demonstrates that this is attainable by a hypernetwork, we conclude that hypernetworks are modular. In contrast, embedding methods are not since, as Thms. 2-3 show, they require a significantly larger primary.

**Related Work**    Hypernetworks, which were first introduced under this name in [11], are networks that generate the weights of a second *primary* network that computes the actual task. The Bayesian formulation of [18] introduces variational inference that involves both the parameter generating network and a primary network. Hypernetworks are especially suited for meta-learning tasks, such as few-shot [1] and continual learning tasks [36], due to the knowledge sharing ability of the weights generating network. Predicting the weights instead of performing backpropagation can lead to efficient neural architecture search [2, 37], and hyperparameter selection [22].

Multiplicative interactions, such as gating, attention layers, hypernetworks, and dynamic convolutions, were shown to strictly extend standard neural networks [15]. However, the current literature has no theoretical guarantees that support the claim that interactions have a clear advantage.

In this work, we take an approximation theory perspective of this problem. For this purpose, as a starting point, we study standard neural networks as function approximators. There were various attempts to understand the capabilities of neural networks as universal function approximators [5, 14]. Multiple extensions of these results [29, 23, 12, 20, 32] quantify tradeoffs between the number of trainable parameters, width and depth of the neural networks as universal approximators. In particular, [29] suggested upper bounds on the size of the neural networks of order $\mathcal{O}(\epsilon^{-n/r})$, where $n$ is the input dimension, $r$ is the order of smoothness of the target functions, and $\epsilon > 0$ is the approximation accuracy. In another contribution, [6] prove a lower bound on the complexity of the class of approximators that matches the upper bound $\Omega(\epsilon^{-n/r})$. However, their analysis assumes that the approximation is robust in some sense (see Sec. 3 for details). In addition, they show that robustness holds when the class of approximators $\mathcal{f} = \{f(\cdot; \theta) \mid \theta \in \Theta_\mathcal{f}\}$ satisfies a certain notion of bi-Lipschitzness. However, as a consequence of this condition, any two equivalent functions

(i.e., $f(\cdot;\theta_1) = f(\cdot;\theta_2)$) must share the same parameterizations (i.e., $\theta_1 = \theta_2$). Unfortunately, this condition is not met for neural networks, as one can compute the same function with neural networks of the same architecture with different parameterizations. In Sec. 3, we show that for certain activation functions and under reasonable conditions, there exists a robust approximator and, therefore, the lower bound of the complexity is $\Omega(\epsilon^{-n/r})$. Since the existence of a continuous selector is also a cornerstone in the proofs of Thms. 2-5, the analysis in [6] is insufficient to prove these results (for example, see the proof sketch of Thm. 4 in Sec. 4.1). In [24, 25, 26] a similar lower bound is shown, but, only for shallow networks.

In an attempt to understand the benefits of locality in convolutional neural networks, [28] shows that when the target function is a hierarchical function, it can be approximated by a hierarchic neural network of smaller complexity, compared to the worst-case complexity for approximating arbitrary functions. In our Thm. 5, we take a similar approach. We show that under standard assumptions in meta-learning, the overall number of trainable parameters in a hypernetwork necessary to approximate the target function is smaller by orders of magnitude, compared to approximating arbitrary functions with neural networks and the embedding method in particular.

## 2    Problem Setup

In various meta-learning settings, we have an unknown target function $y : \mathcal{X} \times \mathcal{I} \to \mathbb{R}$ that we would like to model. Here, $x \in \mathcal{X}$ and $I \in \mathcal{I}$ are two different inputs of $y$. The two inputs have different roles, as the input $I$ is "task" specific and $x$ is independent of the task. Typically, the modeling of $y$ is done in the following manner: $H(x, I) = G(x, E(I)) \approx y(x, I)$, where $E$ is an embedding function and $G$ is a predictor on top of it. The distinction between different embedding methods stems from the architectural relationship between $E$ and $G$. In this work, we compare two task embedding methods: (i) neural embedding methods and (ii) hypernetworks.

A **neural embedding method** is a network of the form $h(x, I; \theta_e, \theta_q) = q(x, e(I; \theta_e); \theta_q)$, consisting of a composition of neural networks $q$ and $e$ parameterized with real-valued vectors $\theta_q \in \Theta_q$ and $\theta_e \in \Theta_e$ (resp.). The term $e(I; \theta_e)$ serves as an embedding of $I$. For two given families $\mathcal{q} := \{q(x, z; \theta_q) \mid \theta_q \in \Theta_q\}$ and $\mathcal{e} := \{e(I; \theta_e) \mid \theta_e \in \Theta_e\}$ of functions, we denote by $\mathcal{E}_{e,q} := \{q(x, e(I; \theta_e); \theta_q) \mid \theta_q \in \Theta_q, \theta_e \in \Theta_e\}$ the embedding method that is formed by them.

A special case of neural embedding methods is the family of the conditional neural processes models [8]. In such processes, $\mathcal{I}$ consists of a set of $d$ images $I = (I_i)_{i=1}^d \in \mathcal{I}$, and the embedding is computed as an average of the embeddings over the batch, $e(I; \theta_e) := \frac{1}{d} \sum_{i=1}^d e(I_i; \theta_e)$.

A **hypernetwork** $h(x, I) = g(x; f(I; \theta_f))$ is a pair of collaborating neural networks, $f : \mathcal{I} \to \Theta_g$ and $g : \mathcal{X} \to \mathbb{R}$, such that for an input $I$, $f$ produces the weights $\theta_I = f(I; \theta_f)$ of $g$, where $\theta_f \in \Theta_f$ consists of the weights of $f$. The function $f(I; \theta_f)$ takes a conditioning input $I$ and returns the parameters $\theta_I \in \Theta_g$ for $g$. The network $g$ takes an input $x$ and returns an output $g(x; \theta_I)$ that depends on both $x$ and the task specific input $I$. In practice, $f$ is typically a large neural network and $g$ is a small neural network.

The entire prediction process for hypernetworks is denoted by $h(x, I; \theta_f)$, and the set of functions $h(x, I; \theta_f)$ that are formed by two families $\mathcal{f} := \{f(I; \theta_f) \mid \theta_f \in \Theta_f\}$ and $\mathcal{g} := \{g(x; \theta_g) \mid \theta_g \in \Theta_g\}$ as a hypernetwork is denoted by $\mathcal{H}_{f,g} := \{g(x; f(I; \theta_f)) \mid \theta_f \in \Theta_f\}$.

### 2.1    Terminology and Notations

We consider $\mathcal{X} = [-1, 1]^{m_1}$ and $\mathcal{I} = [-1, 1]^{m_2}$ and denote, $m := m_1 + m_2$. For a closed set $X \subset \mathbb{R}^n$, we denote by $C^r(X)$ the linear space of all $r$-continuously differentiable functions $h : X \to \mathbb{R}$ on $X$ equipped with the supremum norm $\|h\|_\infty := \max_{x \in X} \|h(x)\|_1$. We denote parametric classes of functions by calligraphic lower letters, e.g., $\mathcal{f} = \{f(\cdot; \theta_f) : \mathbb{R}^m \to \mathbb{R} \mid \theta_f \in \Theta_f\}$. A specific function from the class is denoted by the non-calligraphic lower case version of the letter $f$ or $f(x; \theta_f)$. The notation ";" separates between direct inputs of the function $f$ and its parameters $\theta_f$. Frequently, we will use the notation $f(\cdot; \theta_f)$, to specify a function $f$ and its parameters $\theta_f$ without specifying a concrete input of this function. The set $\Theta_f$ is closed a subset of $\mathbb{R}^{N_f}$ and consists of the various parameterizations of members of $\mathcal{f}$ and $N_f$ is the number of parameters in $\mathcal{f}$, referred to as the complexity of $\mathcal{f}$.

A class of neural networks $f$ is a set of functions of the form:

$$f(x; [\boldsymbol{W}, \boldsymbol{b}]) := W^k \cdot \sigma(W^{k-1} \ldots \sigma(W^1 x + b^1) + b^{k-1}) \tag{1}$$

with weights $W^i \in \mathbb{R}^{h_{i+1} \times h_i}$ and biases $b^i \in \mathbb{R}^{h_{i+1}}$, for some $h_i \in \mathbb{N}$. In addition, $\theta := [\boldsymbol{W}, \boldsymbol{b}]$ accumulates the parameters of the network. The function $\sigma$ is a non-linear activation function, typically ReLU, logistic function, or the hyperbolic tangent.

We define the spectral complexity of a network $f := f(\cdot; [\boldsymbol{W}, \boldsymbol{b}])$ as $\mathcal{C}(f) := \mathcal{C}([\boldsymbol{W}, \boldsymbol{b}]) := L^{k-1} \cdot \prod_{i=1}^{k} \|W^i\|_1$, where $\|W\|_1$ is the induced $L_1$ matrix norm and $L$ is the Lipschitz constant of $\sigma$. In general, $\mathcal{C}(f)$ upper bounds the Lipschitz constant of $f$ (see Lem. 3 in the appendix).

Throughout the paper, we consider the Sobolev space $\mathcal{W}_{r,n}$ as the set of target functions to be approximated. This class consists of $r$-smooth functions of bounded derivatives. Formally, it consists of functions $h : [-1, 1]^n \to \mathbb{R}$ with continuous partial derivatives of orders up to $r$, such that, the Sobolev norm is bounded, $\|h\|_r^s := \|h\|_\infty + \sum_{1 \le |\mathbf{k}|_1 \le r} \|D^{\mathbf{k}} h\|_\infty \le 1$, where $D^{\mathbf{k}}$ denotes the partial derivative indicated by the multi–integer $\mathbf{k} \ge 1$, and $|\mathbf{k}|_1$ is the sum of the components of $\mathbf{k}$. Members of this class are typically the objective of approximation in the literature [29, 24, 23].

In addition, we define the class $\mathcal{P}_{r,w,c}^{k_1,k_2}$ to be the set of functions $h : \mathbb{R}^{k_1} \to \mathbb{R}^{k_2}$ of the form $h(x) = W \cdot P(x)$, where $P : \mathbb{R}^{k_1} \to \mathbb{R}^w$ and $W \in \mathbb{R}^{k_2 \times w}$ is some matrix of the bounded induced $L_1$ norm $\|W\|_1 \le c$. Each output coordinate $P_i$ of $P$ is a member of $\mathcal{W}_{r,k_1}$. The linear transformation on top of these functions serves to enable blowing up the dimension of the produced output. However, the "effective" dimensionality of the output is bounded by $w$. For simplicity, when $k_1$ and $k_2$ are clear from context, we simply denote $\mathcal{P}_{r,w,c} := \mathcal{P}_{r,w,c}^{k_1,k_2}$. We can think of the functions in this set as linear projections of a set of features of size $w$.

**Assumptions** Several assumptions were made to obtain the theoretical results. The first one is not strictly necessary, but significantly reduces the complexity of the proofs: we assume the existence of a unique function $f \in f$ that best approximates a given target function $y$. It is validated empirically in Sec. 5.

**Assumption 1** (Unique Approximation)**.** *Let $f$ be a class of neural networks. Then, for all $y \in \mathbb{Y}$ there is a unique function $f(\cdot; \theta^*) \in f$ that satisfies: $\|f(\cdot; \theta^*) - y\|_\infty = \inf_{\theta \in \Theta_f} \|f(\cdot; \theta) - y\|_\infty$.*

For simplicity, we also assume that the parameters $\theta^*$ of the best approximators are bounded (uniformly, for all $y \in \mathbb{Y}$). The next assumption is intuitive and asserts that for any target function $y$ that is being approximated by a class of neural networks $f$, by adding a neuron to the architecture, one can achieve a strictly better approximation to $y$ or $y$ is already perfectly approximated by $f$.

**Assumption 2.** *Let $f$ be a class of neural networks. Let $y \in \mathbb{Y}$ be some function to be approximated. Let $f'$ be a class of neural networks that resulted by adding a neuron to some hidden layer of $f$. If $y \notin f$ then, $\inf_{\theta \in \Theta_f} \|f(\cdot; \theta) - y\|_\infty > \inf_{\theta \in \Theta_{f'}} \|f(\cdot; \theta) - y\|_\infty$.*

This assumption is validated empirically in Sec. 1.5 of the appendix. In the following lemma, we prove that Assumption 2 holds for shallow networks for the $L_2$ distance instead of $L_\infty$.

**Lemma 1.** *Let $\mathbb{Y} = C([-1, 1]^m)$ be the class of continuous functions $y : [-1, 1]^m \to \mathbb{R}$. Let $f$ be a class of 2-layered neural networks of width $d$ with $\sigma$ activations, where $\sigma$ is either $\tanh$ or sigmoid. Let $y \in \mathbb{Y}$ be some function to be approximated. Let $f'$ be a class of neural networks that is resulted by adding a neuron to the hidden layer of $f$. If $y \notin f$ then, $\inf_{\theta \in \Theta_f} \|f(\cdot; \theta) - y\|_2^2 > \inf_{\theta \in \Theta_{f'}} \|f(\cdot; \theta) - y\|_2^2$. The same holds for $\sigma = ReLU$ when $m = 1$.*

## 3 Degrees of Approximation

We are interested in determining how complex a model ought to be to theoretically guarantee approximation of an unknown target function $y$ up to a given approximation error $\epsilon > 0$. Formally, let $\mathbb{Y}$ be a set of target functions to be approximated. For a set $\mathcal{P}$ of candidate approximators, we measure its ability to approximate $\mathbb{Y}$ as: $d(\mathcal{P}; \mathbb{Y}) := \sup_{y \in \mathbb{Y}} \inf_{p \in \mathcal{P}} \|y - p\|_\infty$. This quantity measures the maximal approximation error for approximating a target function $y \in \mathbb{Y}$ using candidates $p$ from $\mathcal{P}$.

Typical approximation results show that the class $\mathbb{Y} = \mathcal{W}_{r,m}$ can be approximated using classes of neural networks $f$ of sizes $\mathcal{O}(\epsilon^{-m/r})$, where $\epsilon$ is an upper bound on $d(f; \mathbb{Y})$. For instance, in [29]

this property is shown for neural networks with activations $\sigma$ that are infinitely differentiable and not polynomial on any interval; [12] prove this property for ReLU neural networks. We call activation functions with this property *universal*.

**Definition 1** (Universal activation). *An activation function $\sigma$ is universal if for any $r, n \in \mathbb{N}$ and $\epsilon > 0$, there is a class of neural networks $f$ with $\sigma$ activations, of size $\mathcal{O}(\epsilon^{-n/r})$, such that, $d(f; \mathcal{W}_{r,n}) \leq \epsilon$.*

An interesting question is whether this bound is tight. We recall the $N$-width framework of [6] (see also [31]). Let $f$ be a class of functions (not necessarily neural networks) and $S : \mathbb{Y} \to \mathbb{R}^N$ be a continuous mapping between a function $y$ and its approximation, where with $N := N_f$. In this setting, we approximate $y$ using $f(\cdot; S(y))$, where the continuity of $S$ means that the selection of parameters is robust with respect to perturbations in $y$. The nonlinear $N$-width of the compact set $\mathbb{Y} = \mathcal{W}_{r,m}$ is defined as follows:

$$\tilde{d}_N(\mathbb{Y}) := \inf_f \tilde{d}(f; \mathbb{Y}) := \inf_f \inf_S \sup_{y \in \mathbb{Y}} \|f(\cdot; S(y)) - y\|_\infty, \tag{2}$$

where the infimum is taken over classes $f$, such that, $N_f = N$ and $S$ is continuous. Informally, the $N$-width of the class $\mathbb{Y}$ measures the minimal approximation error achievable by a continuous function $S$ that selects approximators $f(\cdot; S(y))$ for the functions $y \in \mathbb{Y}$. As shown by [6], $\tilde{d}_N(\mathbb{Y}) = \Omega(N^{-m/r})$, or alternatively, if there exists $f$, such that, $\tilde{d}(f; \mathbb{Y}) \leq \epsilon$ (i.e., $\tilde{d}_{N_f}(\mathbb{Y}) \leq \epsilon$), then $N_f = \Omega(\epsilon^{-m/r})$. We note that since the $N$-width of $\mathbb{Y}$ is oblivious of the class of approximators $f$ and $d(f; \mathbb{Y}) \leq \tilde{d}(f; \mathbb{Y})$ and, therefore, this analysis does not provide a full solution to this question. Specifically, to answer this question, it requires a nuanced treatment of the considered class of approximators $f$.

In the following theorem, we show that under certain conditions, the lower bound holds, even when removing the assumption that the selection is robust.

**Theorem 1.** *Let $\sigma$ be a piece-wise $C^1(\mathbb{R})$ activation function with $\sigma' \in BV(\mathbb{R})$. Let $f$ be a class of neural networks with $\sigma$ activations. Let $\mathbb{Y} = \mathcal{W}_{r,m}$. Assume that any non-constant $y \in \mathbb{Y}$ is not a member of $f$. Then, if $d(f; \mathbb{Y}) \leq \epsilon$, we have $N_f = \Omega(\epsilon^{-m/r})$.*

All of the proofs are provided in the appendix. The notation $BV(\mathbb{R})$ stands for the set of functions of bounded variation,

$$BV(\mathbb{R}) := \{f \in L^1(\mathbb{R}) \mid \|f\|_{BV} < \infty\} \text{ where, } \|f\|_{BV} := \sup_{\substack{\phi \in C_c^1(\mathbb{R}) \\ \|\phi\|_\infty \leq 1}} \int_\mathbb{R} f(x) \cdot \phi(x) \, \mathrm{d}x \tag{3}$$

We note that a wide variety of activation functions satisfy the conditions of Thm. 1, such as, the clipped ReLU, sigmoid, $\tanh$ and $\arctan$. Informally, to prove this theorem, we show the existence of a "wide" subclass $\mathbb{Y}' \subset \mathbb{Y}$ and a **continuous selector** $S : \mathbb{Y}' \to \Theta_f$, such that, $\exists \alpha > 0 \; \forall y \in \mathbb{Y}' :$ $\|f(\cdot; S(y)) - y\|_\infty \leq \alpha \cdot \inf_{\theta \in \Theta_f} \|f(\cdot; \theta) - y\|_\infty$. The class $\mathbb{Y}'$ is considered wide in terms of $N$-width, i.e., $\tilde{d}_N(\mathbb{Y}') = \Omega(N^{-m/r})$. Therefore, we conclude that $d(f; \mathbb{Y}) \geq d(f; \mathbb{Y}') \geq \frac{1}{\alpha}\tilde{d}(f; \mathbb{Y}') = \Omega(N^{-m/r})$. For further details, see the proof sketches in Secs. 3.2-3.3 of the appendix. Finally, we note that the assumption that any non-constant $y \in \mathbb{Y}$ is not a member of $f$ is rather technical. For a relaxed, for general version of it, see Lem. 18 in the appendix.

## 4 Expressivity of Hypernetworks

Using Thm. 1, the expressive power of hypernetworks is demonstrated. In the first part, we compare the complexities of $g$ and $q$. We show that when letting $e$ and $f$ be large enough, one can approximate it using a hypernetwork where $g$ is smaller than $q$ by orders of magnitude. In the second part, we show that under typical assumptions on $y$, one can approximate $y$ using a hypernetwork with overall much fewer parameters than the number of parameters required for a neural embedding method. It is worth mentioning that our results scale to the multi-dimensional case. In this case, if the output dimension is constant, we get the exact same bounds.

### 4.1 Comparing the complexities of $q$ and $g$

We recall that for an arbitrary $r$-smooth function $y \in \mathcal{W}_{r,n}$, the complexity for approximating it is $\mathcal{O}(\epsilon^{-n/r})$. We show that hypernetwork models can effectively learn a different function for each input

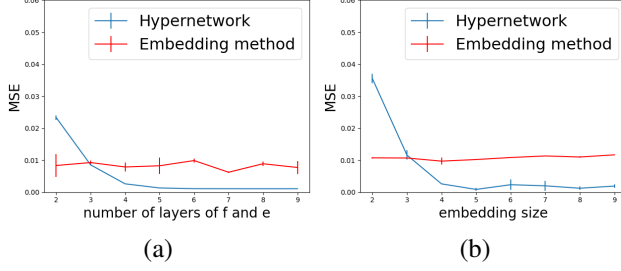

Figure 1: (a) MSE error obtained by hypernetworks and the embedding method with varying number of layers (x-axis). Synthetic target functions $y(x, I) = \langle x, h(I) \rangle$, for neural network $h$. (b) Varying the embedding layer to be 100/1000 (depending on the method) times the x-axis. error bars are SD over 100 repetitions.

instance $I$. Specifically, a hypernetwork is able to capture a separate approximator $h_I = g(\cdot; f(I; \theta_f))$ for each $y_I$ that has a minimal complexity $\mathcal{O}(\epsilon^{-m_1/r})$. On the other hand, we show that for a smoothness order of $r = 1$, under certain constraints, when applying an embedding method, it is impossible to provide a separate approximator $h_I = q(\cdot, e(I; \theta_e); \theta_q)$ of complexity $\mathcal{O}(\epsilon^{-m_1})$. Therefore, the embedding method does not enjoy the same modular properties of hypernetworks.

The following result demonstrates that the complexity of the main-network $q$ in any embedding method has to be of non-optimal complexity. As we show, it holds regardless of the size of $e$, as long as the functions $e \in \mathfrak{e}$ are of bounded Lipschitzness.

**Theorem 2.** *Let $\sigma$ be a universal, piece-wise $C^1(\mathbb{R})$ activation function with $\sigma' \in BV(\mathbb{R})$ and $\sigma(0) = 0$. Let $\mathcal{E}_{e,q}$ be a neural embedding method. Assume that $\mathfrak{e}$ is a class of continuously differentiable neural network $e$ with zero biases, output dimension $k = \mathcal{O}(1)$ and $\mathcal{C}(e) \le \ell_1$ and $\mathfrak{q}$ is a class of neural networks $q$ with $\sigma$ activations and $\mathcal{C}(q) \le \ell_2$. Let $\mathbb{Y} := \mathcal{W}_{1,m}$. Assume that any non-constant $y \in \mathbb{Y}$ cannot be represented as a neural network with $\sigma$ activations. If the embedding method achieves error $d(\mathcal{E}_{e,q}, \mathbb{Y}) \le \epsilon$, then, the complexity of $\mathfrak{q}$ is: $N_q = \Omega\left(\epsilon^{-(m_1 + m_2)}\right)$.*

The following theorem extends Thm. 2 to the case where the output dimension of $e$ depends on $\epsilon$. In this case, the parameter complexity is also non-optimal.

**Theorem 3.** *In the setting of Thm. 2, except $k$ is not necessarily $\mathcal{O}(1)$. Assume that the first layer of any $q \in \mathfrak{q}$ is bounded $\|W^1\|_1 \le c$, for some constant $c > 0$. If the embedding method achieves error $d(\mathcal{E}_{e,q}, \mathbb{Y}) \le \epsilon$, then, the complexity of $\mathfrak{q}$ is: $N_q = \Omega\left(\epsilon^{-\min(m, 2m_1)}\right)$.*

The results in Thms. 2-3 are limited to $r = 1$, which in the context of the Sobolev space $r = 1$ means bounded, Lipschitz and continuously differentiable functions. The ability to approximate these functions is studied extensively in the literature [23, 12]. Extending the results for $r > 1$ is possible but necessitates the introduction of spectral complexities that correspond to higher-order derivatives.

The following theorem shows that for any function $y \in \mathcal{W}_{r,m}$, there is a large enough hypernetwork, that maps between $I$ and an approximator of $y_I$ of optimal complexity.

**Theorem 4** (Modularity of Hypernetworks). *Let $\sigma$ be as in Thm. 2. Let $y \in \mathbb{Y} = \mathcal{W}_{r,m}$ be a function, such that, $y_I$ cannot be represented as a neural network with $\sigma$ activations for all $I \in \mathcal{I}$. Then, there is a class, $\mathfrak{q}$, of neural networks with $\sigma$ activations and a network $f(I; \theta_f)$ with ReLU activations, such that, $h(x, I) = g(x; f(I; \theta_f))$ achieves error $\le \epsilon$ in approximating $y$ and $N_g = \mathcal{O}\left(\epsilon^{-m_1/r}\right)$.*

Recall that Thm. 1 shows that the minimal complexity for approximating each individual smooth target function $y_I$ is $\mathcal{O}(\epsilon^{-m_1/r})$. Besides, Thm. 4 shows that this level of fitting is attainable by a hypernetwork for all $y_I$. Therefore, we conclude that hypernetworks are modular. On the other hand, from Thms. 2-3 we conclude that this is not the case for the embedding method.

When comparing the results in Thms. 2, 3 and 4 in the case of $r = 1$, we notice that in the hypernetworks case, $\mathfrak{q}$ can be of complexity $\mathcal{O}(\epsilon^{-m_1})$ in order to achieve approximation error $\le \epsilon$. On the other hand, for the embedding method case, the complexity of the primary-network $q$ is at least $\Omega(\epsilon^{-(m_1 + m_2)})$ when the embedding dimension is of constant size and at least $\Omega\left(\epsilon^{-\min(m, 2m_1)}\right)$ when it is unbounded to achieve approximation error $\le \epsilon$. In both cases, the primary network of the embedding method is larger by orders of magnitude than the primary network of the hypernetwork.

Note that the embedding method can be viewed as a simple hypernetwork, where only the biases of the first layer of $g$ are given by $f$. Therefore, the above results show that the modular property of hypernetworks, which enables $g$ to be of small complexity, emerges only when letting $f$ produce the

whole set of weights of $g$. We note that this kind of emulation is not symmetric, as it is impossible to emulate a hypernetwork with the embedding method as it is bound to a specific structure defined by $q$ being a neural network (as in Eq. 1) that takes the concatenation of $x$ and $e(I; \theta_e)$ as its input.

**Proof sketch of Thm. 4** Informally, the theorem follows from three main arguments: **(i)** we treat $y(x, I)$ as a class of functions $\mathcal{Y} := \{y_I\}_{I \in \mathcal{I}}$ and take a class $g$ of neural networks of size $\mathcal{O}(\epsilon^{-m_2/r})$, that achieves $d(g; \mathcal{Y}) \le \epsilon$, **(ii)** we prove the existence of a continuous selector for $\mathcal{Y}$ within $g$ and **(iii)** we draw a correspondence between the continuous selector and modeling $y$ using a hypernetwork.

We want to show the existence of a class $g$ of size $\mathcal{O}(\epsilon^{-m_2/r})$ and a network $f(I; \theta_f)$, such that,

$$\sup_I \|g(\cdot; f(I; \theta_f)) - y_I\|_\infty \le 3 \sup_I \inf_{\theta_g} \|g(\cdot; \theta_g) - y_I\|_\infty \le 3\epsilon \tag{4}$$

We note that this expression is very similar to a robust approximation of the class $\mathcal{Y}$, except the selector $S(y_I)$ is replaced with a network $f(I; \theta_f)$. Since $\sigma$ is universal, there exists an architecture $g$ of size $\mathcal{O}(\epsilon^{-m_1/r})$, such that, $d(g; \mathcal{Y}) \le \epsilon$. In addition, we prove the existence of a continuous selector $S : \mathcal{Y} \to \Theta_g$, i.e., $\sup_I \|g(\cdot; S(y_I)) - y_I\|_\infty \le 2d(g; \mathcal{Y}) \le 2\epsilon$.

As a next step, we replace $S$ with a neural network $f(I; \theta_f)$. Since $I \mapsto y_I$ is a continuous function, the function $\hat{S}(I) := S(y_I)$ is continuous as well. Furthermore, as we show, $g$ is uniformly continuous with respect to both $x$ and $\theta_g$, and therefore, by ensuring that $\inf_{\theta_f} \|f(\cdot; \theta_f) - S(\cdot)\|_\infty$ is small enough, we can guarantee that $\inf_{\theta_f} \sup_I \|g(\cdot; f(I; \theta_f)) - g(\cdot; \hat{S}(I))\|_\infty \le \epsilon$. Indeed, by [12], if $f$ is a class of large enough ReLU neural networks, we can ensure that $\inf_{\theta_f} \|f(\cdot; \theta_f) - S(\cdot)\|_\infty$ is as small as we wish. Hence, by the triangle inequality, we have: $\inf_{\theta_f} \sup_I \|g(\cdot; f(I; \theta_f)) - y_I\|_\infty \le 3\epsilon$.

## 4.2 Parameter Complexity of Meta-Networks

As discussed in Sec. 4.1, there exists a selection function $S : \mathcal{I} \to \Theta_g$ that takes $I$ and returns parameters of $g$, such that, $g(\cdot; S(I))$ well approximate $y_I$. In common practical scenarios, the typical assumption regarding the selection function $S(I)$ is that it takes the form $W \cdot h$, for some continuous function $h : \mathcal{I} \to \mathbb{R}^w$ for some relatively small $w > 0$ and $W$ is a linear mapping [34, 22, 3, 21]. In this section, we show that for functions $y$ with a continuous selector $S$ of this type, the complexity of the function $f$ can be reduced from $\mathcal{O}(\epsilon^{-m/r})$ to $\mathcal{O}(\epsilon^{-m_2/r} + \epsilon^{-m_1/r})$.

**Theorem 5.** *Let $\sigma$ be a in Thm. 2. Let $g$ be a class of neural networks with $\sigma$ activations. Let $y \in \mathbb{Y} := \mathcal{W}_{r,m}$ be a target function. Assume that there is a continuous selector $S \in \mathcal{P}_{r,w,c}$ for the class $\{y_I\}_{I \in \mathcal{I}}$ within $g$. Then, there is a hypernetwork $h(x, I) = g(x; f(I; \theta_f))$ that achieves error $\le \epsilon$ in approximating $y$, such that: $N_f = \mathcal{O}(w^{1+m_2/r} \cdot \epsilon^{-m_2/r} + w \cdot N_g) = \mathcal{O}(\epsilon^{-m_2/r} + \epsilon^{-m_1/r})$.*

We note that the number of trainable parameters in a hypernetwork is measured by $N_f$. By Thm. 1, the number of trainable parameters in a neural network is $\Omega(\epsilon^{-(m_1+m_2)/r})$ in order to be able to approximate any function $y \in \mathcal{W}_{r,m}$. Thm. 3 shows that in the case of the common hypernetwork structure, the number of trainable parameters of the hypernetwork is reduced to $\mathcal{O}(\epsilon^{-m_2/r} + \epsilon^{-m_1/r})$. While for embedding methods, where the total number of parameters combines those of both $q$ and $e$, it is evident that the overall number of trainable parameters is $\Omega(\epsilon^{-(m_1+m_2)/r})$. In particular, when equating the number of trainable parameters of a hypernetwork with the size of an embedding method, the hypernetworks' approximation error is significantly lower. This kind of stronger rates of realizability is typically associated with an enhanced generalization performance [33].

## 5 Experiments

**Validating Assumption 1** Informally, Assumption 1 claims that for any target function $y \in \mathbb{Y}$, and class $f$ of neural networks with an activation function $\sigma$, there is a unique global approximator $f^* \in f$, such that, $f^* \in \arg\inf_{f \in f} \|f - y\|_\infty$. To empirically validate the assumption, we take a high complexity target function $y$ and approximate it using two neural network approximators $f_1$ and $f_2$ of the same architecture $f$. The goal is to show that when $f_1$ and $f_2$ are best approximators of $y$ within $f$, then, they have similar input-output relations, regardless of approximation error.

Three input spaces are considered: (i) the CIFAR10 dataset, (ii) the MNIST dataset and (iii) the set $[-1, 1]^{28 \times 28}$. The functions $f_1$ and $f_2$ are shallow ReLU MLP neural networks with 100 hidden

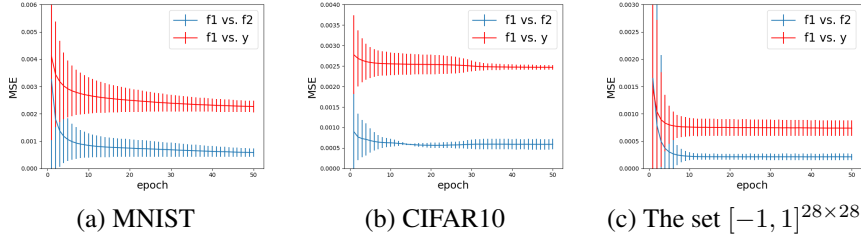

| (a) MNIST | (b) CIFAR10 | (c) The set $[-1, 1]^{28 \times 28}$ |

Figure 2: **Validating Assumption 1.** MSE between $f_1$ and $f_2$ (blue), and between $f_1$ and $y$ (red), when approximating $y$, as a function of epoch.

neurons and 10 output neurons. The target function $y$ is a convolutional neural network of the form:

$$y(x) = \text{fc}_1 \circ \text{ReLU} \circ \text{conv}_2 \circ \text{ReLU} \circ \text{conv}_1(x) \tag{5}$$

where $\text{conv}_1$ ($\text{conv}_2$) is a convolutional layer with 1 or 3 (20) input channels, 20 (50) output channels, kernel size 10 and stride 2 and $\text{fc}_1$ with 10 outputs.

To study the convergence between $f_1$ and $f_2$, we train them independently to minimize the MSE loss to match the output of $y$ on random samples from the input space. The training was done using the SGD method with a learning rate $\mu = 0.01$ and momentum $\gamma = 0.5$, for 50 epochs. We initialized $f_1$ and $f_2$ using different initializations.

In Fig. 2 we observe that the distance between $f_1$ and $f_2$ tends to be significantly smaller than their distances from $y$. Therefore, we conclude that regardless of the approximation error of $y$ within $\ell$, any two best approximators $f_1, f_2 \in \ell$ of $y$ are identical.

**Synthetic Experiments**  We experimented with the following class of target functions. The dimensions of $x$ and $I$ are denoted by $d_x$ and $d_I$ (resp.). The target functions is of the form $y(x, I) := \langle x, h(I) \rangle$, $h$ is a three-layers fully-connected sigmoid neural network. See the appendix for further details and experiments with two additional classes of target functions.

*Varying the number of layers*  To compare between the two models, we took the primary-networks $g$ and $q$ to be neural networks with two layers of dimensions $d_{\text{in}} \to 10 \to 1$ and ReLU activation within the hidden layer. The input dimension of $g$ is $d_{\text{in}} = d_x = 10^3$ and for $q$ is $d_{\text{in}} = d_x + E = 10^3 + 10^4$. In addition, $f$ and $e$ are neural networks with $k = 2, \dots, 9$ layers, each layer of width 100. The output dimension of $e$ is $E = 10^4$. In this case, the size of $q$ is $N_q = 10^4 + 10E + 10$, which is larger than the size of $g$, $N_g = 10^4 + 10$. The sizes of $f$ and $e$ are $N_f = 10^5 + 10^4 \cdot (k - 2) + 10^2 \cdot N_g$ and $N_e = 10^5 + 10^4 \cdot (k - 2) + 10^6$, which are both of order $10^6$.

We compared the MSE losses at the test time of the hypernetwork and the embedding method in approximating the target function $y$. The training was done over 30000 samples $(x, I, y(x, I))$, with $x$ and $I$ taken from a standard normal distribution. The samples are divided into batches of size 200 and the optimization is done using the SGD method with a learning rate $\mu = 0.01$.

As can be seen in Fig. 1(a), when the number of layers of $f$ and $e$ are $\geq 3$, the hypernetwork model outperforms the embedding method. It is also evident that the approximation error of hypernetworks improves, as long as we increase the number of layers of $f$. This is in contrast to the case of the embedding method, the approximation error does not improve when increasing $e$'s number of layers. These results are very much in line with the theorems in Sec. 4.2. As can be seen in Thms. 2 and 4, when fixing the sizes of $g$ and $q$, while letting $f$ and $e$ be as large as we wish we can achieve a much better approximation with the hypernetwork model.

*Varying the embedding dimension*  Next, we investigate the effect of varying the embedding dimension in both models to be $10^2 i$, for $i \in [8]$. In this experiment, $d_x = d_I = 100$, the primary-networks $g$ and $q$ are set to be ReLU networks with two layers of dimensions $d_{\text{in}} \to 10 \to 1$. The input dimension of $g$ is $d_{\text{in}} = d_x = 100$ and for $q$ is $d_{\text{in}} = d_x + 100i$. The functions $f$ and $e$ are fully connected networks with three layers. The dimensions of $f$ are $10^2 \to 10^2 \to 10^2 i \to N_g$ and the dimensions of $e$ are $10^2 \to 10^2 \to 10^2 \to 10^3 i$. The overall size of $g$ is $N_g = 1010$ which is smaller than the size of $q$, $N_q = 10^4(i + 1) + 10$. The size of $f$ is $N_f = 10^4 + 10^4 i + 10^5 i$ and the size of $e$ is $N_e = 2 \cdot 10^4 + 10^5 i$ which are both $\approx 10^5 i$.

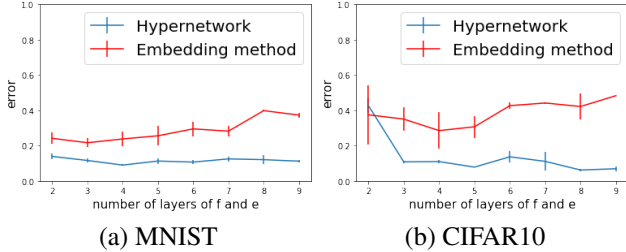

| (a) MNIST | (b) CIFAR10 |

Figure 3: **Predicting image rotations.** (a-b) The error obtained by hypernetworks and the embedding method with a varying number of layers (x-axis).

As can be seen from Fig. 1(b), the performance of the embedding method does not improve when increasing the embedding dimension. Also, the overall performance is much worse than the performance of hypernetworks with deeper or wider $f$. This result verifies the claim in Thm. 3 that by increasing the embedding dimension the embedding model is unable to achieve the same rate of approximation as the hypernetwork model.

**Experiments on Real-world Datasets**     To validate the prediction in Sec. 4.1, we experimented with comparing the ability of hypernetworks and embedding methods of similar complexities in approximating the target function. We experimented with the MNIST [19] and CIFAR10 datasets [17] on two self-supervised learning tasks: predicting image rotations, described below and image colorization (Sec. 1.3 in the appendix). For image rotation, the target functions are $y(x, I)$, where $I$ is a sample from the dataset and $x$ is a rotated version of it with a random angle $\alpha$, which is a self-supervised task [13, 9, 7, 10]. The function $y$ is the closest value to $\alpha/360$ within $\{\alpha_i = 30i/360 \mid i = 0, \ldots, 11\}$. The inputs $x$ and $I$ are flattened and their dimensions are $d_x = d_I = h^2 c$, where $h, c$ are the height and number of channels of the images.

*Varying the number of layers*     In this case, the primary-networks $g$ and $q$ are fully connected. The input dimension of $g$ is $d_{\text{in}} = d_x$ and of $q$ is $d_{\text{in}} = d_x + N_g = 11h^2 c + 10$. The functions $f$ and $e$ are ReLU neural networks with a varying number of layers $k = 2, \ldots, 9$. Their input dimensions are $d_I$ and each hidden layer in $e$ and $f$ is of dimension $d$. We took $d = 50$ for MNIST and $d = 100$ for CIFAR10. The output dimensions of $e$ and $f$ are $10h^2 c + 10$. In this case, the numbers of parameters and output dimensions of $e$ and $f$ are the same, since they share the same architecture. In addition, the number of parameters in $g$ is $N_g = 10h^2 c + 10$, while the number of parameters in $q$ is $N_q = 10(11h^2 c + 10) + 10 \approx 10 N_g$.

We compare the classification errors over the test data. The networks are trained with the negative log loss for 10 epochs using SGD with a learning rate of $\mu = 0.01$. We did not apply any regularization or normalization on the two models to minimize the influence of hyperparameters on the comparison.

As can be seen in Fig. 3, the hypernetwork outperforms the embedding method by a wide margin. In contrast to the embedding method, the hypernetwork's performance improves when increasing its depth. For additional experiments on studying the effect of the embedding dimension, see Sec. 1.2 in the appendix. Finally, since the learning rate is the only hyperparameter in the optimization process, we conducted a sensitivity test, showing that the results are consistent when varying the learning rate (see Sec. 1.4 in the appendix).

## 6   Conclusions

We aim to understand the success of hypernetworks from a theoretical standpoint and compared the complexity of hypernetworks and embedding methods in terms of the number of trainable parameters. To achieve error $\leq \epsilon$ when modeling a function $y(x, I)$ using hypernetworks, the primary-network can be selected to be of a much smaller family of networks than the primary-network of an embedding method. This result manifests the ability of hypernetworks to effectively learn distinct functions for each $y_I$ separately. While our analysis points to the existence of modularity in hypernetworks, it does not mean that this modularity is achievable through SGD optimization. However, our experiments as well as the successful application of this technology in practice, specifically using a large $f$ and a small $g$, indicate that this is indeed the case, and the optimization methods are likely to converge to modular solutions.

## Broader Impact

Understanding modular models, in which learning is replaced by meta-learning, can lead to an ease in which models are designed and combined at an abstract level. This way, deep learning technology can be made more accessible. Beyond that, this work falls under the category of basic research and does not seem to have particular societal or ethical implications.

## Acknowledgements and Funding Disclosure

This project has received funding from the European Research Council (ERC) under the European Union's Horizon 2020 research and innovation programme (grant ERC CoG 725974). The contribution of Tomer Galanti is part of Ph.D. thesis research conducted at Tel Aviv University.

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
