[Supplementary Material]

# Appendix: On the Modularity of Hypernetworks

**Tomer Galanti**
School of Computer Science
Tel Aviv University
Tel Aviv, Israel
tomerga2@tauex.tau.ac.il

**Lior Wolf**
Facebook AI Research (FAIR) &
School of Computer Science
Tel Aviv University
Tel Aviv, Israel
wolf@fb.com

## 1 Additional Experiments

### 1.1 Synthetic Experiments

As an additional experiment, we repeated the same experiment (i.e., varying the number of layers of $f$ and $e$ or the embedding dimension) in Sec. 5 with two different classes of target functions (type II and III). The experiments with Type I functions are presented in the main text.

**Type I**   The target functions is of the form $y(x, I) := \langle x, h(I) \rangle$. Here, $h$ is a three-layers fully-connected neural network of dimensions $d_I \rightarrow 300 \rightarrow 300 \rightarrow 10^3$ and applies sigmoid activations within the two hidden layers and softmax on top of the network. The reason we apply softmax on top of the network is to restrict its output to be bounded.

**Type II**   The second group of functions consists of randomly initialized fully connected neural networks $y(x, I)$. The neural network has four layers of dimensions $(d_x + d_I) \rightarrow 100 \rightarrow 50 \rightarrow 50 \rightarrow 1$ and applies ELU activations.

**Type III**   The second type of target functions $y(x, I) := h(x \odot I)$ consists of fully-connected neural network applied on top of the element-wise multiplication between $x$ and $I$. The neural network consists of four layers of dimensions $d_I \rightarrow 100 \rightarrow 100 \rightarrow 50 \rightarrow 1$ and applies ELU activations. The third type of target functions is of the form $y(x, I) := \langle x, h(I) \rangle$. Here, $h$ is a three-layers fully-connected neural network of dimensions $d_I \rightarrow 300 \rightarrow 300 \rightarrow 1000$ and applies sigmoid activations within the two hidden layers and softmax on top of the network. The reason we apply softmax on top of the network is to restrict its output to be bounded.

In all of the experiments, the weights of $y$ are set using the He uniform initialization [11].

In Fig. 1, we plot the results for varying the number of layers/embedding dimensions of hypernetworks and embedding methods. As can be seen, the performance of hypernetworks improves as a result of increasing the number of layers, despite the embedding method. On the other hand, for both models, increasing the embedding dimension seems ineffective.

### 1.2 Predicting Image Rotations

As an additional experiment on predicting image rotations, we studied the effect of the embedding dimension on the performance of the embedding method, we varied the embedding dimension $E_i = 10^4 i$ for $i \in [8]$. The primary-network $q$ has dimensions $d_{\text{in}} \rightarrow 10 \rightarrow 12$ with $d_{\text{in}} = d_I + E_i$ and the embedding network $e$ has architecture $d_x \rightarrow 100 \rightarrow E_i$. We compared the performance to a hypernetwork with $g$ of architecture $d_x \rightarrow 10 \rightarrow 12$ and $f$ of architecture $d_I \rightarrow 100 \rightarrow N_q$. We note that $q$ is larger than $g$, the embedding dimension $E_i$ exceeds $N_q = 30840$ for any $i > 3$ and therefore, $e$ is of larger size than $f$ for $i > 3$.

Figure 1: (a-b) The error obtained by hypernetworks and the embedding method with varying number of layers (x-axis). The MSE (y-axis) is computed between the learned function and the target function at test time. The blue curve stands for the performance of the hypernetwork model and the red one for the neural embedding method. (a) Target functions of neural network type, (b) Functions of the form $y(x, I) = h(x \odot I)$, where $h$ is a neural network.(d-e) Measuring the performance for the same three target functions when varying the size of the embedding layer to be 100/1000 (depending on the method) times the value on the x-axis. The error bars depict the variance across 100 repetitions of the experiment.

Figure 2: **Predicting image rotations.** varying the embedding dimension of the embedding method to be $10^4$ times the value of the x-axis, compared to the results of hypernetworks. The error bars depict the variance across 100 repetitions of the experiment.

As can be seen in Fig. 2, the hypernetwork outperforms the embedding method by a wide margin and the performance of the embedding method does not improve when increasing its embedding dimension.

## 1.3 Image Colorization

The second type of target functions are $y(x, I)$, where $I$ is a sample gray-scaled version of an image $\hat{I}$ from the dataset and $x = (i_1, i_2)$ is a tuple of coordinates, specifying a certain pixel in the image $I$. The function $y(x, I)$ returns the RGB values of $\hat{I}$ in the pixel $x = (i_1, i_2)$ (normalized between $[-1, 1]$). For this self-supervised task we employ CIFAR10 dataset, since the MNIST has grayscale images.

For the purpose of comparison, we considered the following setting. The inputs of the networks are $x' = (i_1, i_2) \| (i_1^k + i_2, i_2^k + i_1, i_1^k - i_2, i_2^k - i_1)_{k=0}^{9}$ and a flattened version of the gray-scaled image $I$ of dimensions $d_{x'} = 42$ and $d_I = 1024$. The functions $f$ and $e$ are fully connected neural networks of the same architecture with a varying number of layers $k = 2, \ldots, 7$. Their input dimension is

$d_I$, each hidden layer is of dimension $100$ and their output dimensions are $450$. We took primary networks $g$ and $q$ to be fully connected neural networks with two layers $d_{\text{in}} \rightarrow 10 \rightarrow 3$ and ELU activations within their hidden layers. For the hypernetwork case, we have: $d_{\text{in}} = 42$ and for the embedding method $d_{\text{in}} = 42 + 450 = 492$, since the input of $q$ is a concatenation of $x'$ (of dimension $42$) and $e(I)$ which is of dimension $450$.

The overall number of trainable parameters in $e$ and $f$ is the same, as they share the same architecture. The number of trainable parameters in $q$ is $492 \cdot 10 + 10 \cdot 3 = 4950$ and in $g$ is $42 \cdot 10 + 10 \cdot 3 = 450$. Therefore, the embedding method is provided with a larger number of trainable parameters as $q$ is $10$ times larger than $g$. The comparison is depicted in Fig. 3. As can be seen, the results of hypernetworks outperform the embedding method by a large margin, and the results improve when increasing the number of layers.

Figure 3: **Colorization.** The error obtained by hypernetworks and the embedding method with varying number of layers (x-axis). The error rate (y-axis) is computed between the learned function and the target function at test time. The blue curve stands for the performance of the hypernetwork model and the red one for the neural embedding method.

## 1.4 Sensitivity Experiment

(a)                                         (b)

Figure 4: **Comparing the performance of a hypernetwork and the embedding method when varying the learning rate.** The x-axis stands for the value of the learning rate and the y-axis stands for the averaged accuracy rate at test time. (a) Results on MNIST and (b) Results on CIFAR10.

In the rotations prediction experiment in Sec. 5, we did not apply any regularization or normalization on the two models to minimize the number of hyperparameters. Therefore, the only hyperparameter we used during the experiment is the learning rate. We conducted a hyperparameter sensitivity test for the learning rate. We compared the two models in the configuration of Sec. 5 when fixing the depths of $f$ and $e$ to be $4$ and varying the learning rate. As can be seen in Fig. 4, the hypernetwork outperforms the baseline for every learning rate in which the networks provide non-trivial error rates.

## 1.5 Validating Assumption 2

To empirically justify Assumption 2, we trained shallow neural networks on MNIST and Fashion MNIST classification with a varying number of hidden neurons. The optimization was done using the MSE loss, where the labels are cast into one-hot encoding. The network is trained using Adadelta with a learning rate of $\mu = 1.0$ and batch size $64$ for $2$ epochs. As can be seen in Fig. 5, the MSE

Figure 5: **Validating Assumption 2.** The MSE loss at test time strictly decreases when increasing the number of hidden neurons.

loss strictly decreases when increasing the number of hidden neurons. This is true for a variety of activation functions.

## 2 Preliminaries

### 2.1 Identifiability

Neural network identifiability is the property in which the input-output map realized by a feed-forward neural network with respect to a given activation function uniquely specifies the network architecture, weights, and biases of the neural network up to neural network isomorphisms (i.e., re-ordering the neurons in the hidden layers). Several publications investigate this property. For instance, [2, 17] show that shallow neural networks are identifiable. The main result of [8] considers feed-forward neural networks with the tanh activation functions are shows that these are identifiable when the networks satisfy certain "genericity assumptions". In [18] it is shown that for a wide class of activation functions, one can find an arbitrarily close function that induces identifiability (see Lem. 1). Throughout the proofs of our Thm. 1, we make use of this last result in order to construct a robust approximator for the target functions of interest.

We recall the terminology of identifiability from [8, 18].

**Definition 1** (Identifiability). *A class $f = \{f(\cdot; \theta_f) : A \to B \mid \theta_f \in \Theta_f\}$ is identifiable up to (invariance) continuous functions $\Pi = \{\pi : \Theta_f \to \Theta_f\}$, if*

$$f(\cdot; \theta_f) \equiv_A f(\cdot; \theta'_f) \iff \exists \pi \in \Pi \text{ s.t } \theta'_f = \pi(\theta_f) \tag{1}$$

*where the equivalence $\equiv_A$ is equality for all $x \in A$.*

A special case of identifiability is identifiability up to isomorphisms. Informally, we say that two neural networks are isomorphic if they share the same architecture and are equivalent up to permuting the neurons in each layer (excluding the input and output layers).

**Definition 2** (Isomorphism). *Let $f$ be a class of neural networks. Two neural networks $f(x; [\mathbf{W}, \mathbf{b}])$ and $f(x; [\mathbf{V}, \mathbf{d}])$ of the same class $f$ are isomorphic if there are permutations $\{\gamma_i : [h_i] \to [h_i]\}_{i=1}^{k+1}$, such that,*

1. *$\gamma_1$ and $\gamma_{k+1}$ are the identity permutations.*

2. *For all $i \in [k]$, $j \in [h_{i+1}]$ and $l \in [h_i]$, we have: $V_{j,l}^i = W_{\gamma_{i+1}(j),\gamma_i(l)}^i$ and $d_j^i = b_{\gamma_{i+1}(j)}^i$.*

*An isomorphism $\pi$ is specified by permutation functions $\gamma_1, \ldots, \gamma_{k+1}$ that satisfy conditions (1) and (2). For a given neural network $f(x; [\mathbf{W}, \mathbf{b}])$ and isomorphism $\pi$, we denote by $\pi \circ [\mathbf{W}, \mathbf{b}]$ the parameters of a neural network produced by the isomorphism $\pi$.*

As noted by [8, 18], for a given class of neural networks, $f$, there are several ways to construct pairs of non-isomorphic neural networks that are equivalent as functions.

In the first approach, suppose that we have a neural network with depth $k \geq 2$, and there exist indices $i, j_1, j_2$ with $1 \leq i \leq k-1$ and $1 \leq j_1 < j_2 \leq h_{i+1}$, such that, $b_{j_1}^i = b_{j_2}^i$ and $W_{j_1,t}^i = W_{j_2,t}^i$ for all $t \in [h_i]$. Then, if we construct a second neural network that shares the same weights and biases, except replacing $W_{1,j_1}^{i+1}$ and $W_{1,j_2}^{i+1}$ with a pair $\tilde{W}_{1,j_1}^{i+1}$ and $\tilde{W}_{1,j_2}^{i+1}$, such that, $\tilde{W}_{1,j_1}^{i+1} + \tilde{W}_{1,j_2}^{i+1} = W_{1,j_1}^{i+1} + W_{1,j_2}^{i+1}$. Then, the two neural networks are equivalent, regardless of the activation function. The $j_1$ and $j_2$ neurons in the $i$'th layer are called clones and are defined formally in the following manner.

**Definition 3** (No-clones condition). *Let class of neural networks $f$. Let $f(x; [\mathbf{W}, \mathbf{b}]) \in f$ be a neural network. We say that $f$ has clone neurons if there are: $i \in [k]$, $j_1 \neq j_2 \in [h_{i+1}]$, such that:*

$$(b_{j_1}^i, W_{j_1,1}^i, \ldots, W_{j_1,h_i}^i) = (b_{j_2}^i, W_{j_2,1}^i, \ldots, W_{j_2,h_i}^i) \tag{2}$$

*If $f$ does not have a clone, we say that $f$ satisfies the no-clones condition.*

A different setting in which uniqueness up to isomorphism is broken, results when taking a neural network that has a "zero" neuron. Suppose that we have a neural network with depth $k \geq 2$, and there exist indices $i, j$ with $1 \leq i \leq k-1$ and $1 \leq j \leq h_{i+1}$, such that, $W_{j,t}^i = 0$ for all $t \in [h_i]$ or $W_{t,j}^{i+1} = 0$ for all $t \in [h_{i+2}]$. In the first case, one can replace any $W_{1,j}^{i+1}$ with any number $\tilde{W}_{1,j}^{i+1}$ if $\sigma(b_{i,j}) = 0$ to get a non-isomorphic equivalent neural network. In the other case, one can replace $W_{j,1}^i$ with any number $\tilde{W}_{j,1}^{i+1}$ to get non-isomorphic equivalent neural network.

**Definition 4** (Minimality). *Let $f(x; [\mathbf{W}, \mathbf{b}])$ be a neural network. We say that $f$ is minimal, if for all $i \in [k]$, each matrix $W^i$ has no identically zero row or an identically zero column.*

A normal neural network satisfies both minimality and the no-clones condition.

**Definition 5** (Normal neural network). *Let $f(x; [\mathbf{W}, \mathbf{b}])$ be a neural network. We say that $f$ is normal, if it has no-clones and is minimal. The set of normal neural networks within $f$ is denoted by $f_n$.*

An interesting question regarding identifiability is whether a given activation $\sigma : \mathbb{R} \to \mathbb{R}$ function implies the identifiability property of any class of normal neural networks $f_n$ with the given activation function are equivalent up to isomorphisms. An activation function of this kind will be called *identifiability inducing*. It has been shown by [8] that the tanh is identifiability inducing up to additional restrictions on the weights. In [17] and in [2] they show that shallow neural networks are identifiable.

**Definition 6** (Identifiability inducing activation). *Let $\sigma : \mathbb{R} \to \mathbb{R}$ be an activation function. We say that $\sigma$ is identifiability inducing if for any class of neural networks $f$ with $\sigma$ activations, we have: $f(\cdot; \theta_1) = f(\cdot; \theta_2) \in f_n$ if and only if they are isomorphic.*

The following theorem by [18] shows that any piece-wise $C^1(\mathbb{R})$ activation function $\sigma$ with $\sigma' \in BV(\mathbb{R})$ can be approximated by an identifiability inducing activation function $\rho$.

**Lemma 1** ([18]). *Let $\sigma : \mathbb{R} \to \mathbb{R}$ be a piece-wise $C^1(\mathbb{R})$ with $\sigma' \in BV(\mathbb{R})$ and let $\epsilon > 0$. Then, there exists a meromorphic function $\rho : D \to \mathbb{C}$, $\mathbb{R} \subset D$, $\rho(\mathbb{R}) \subset \mathbb{R}$, such that, $\|\sigma - \rho\|_\infty < \epsilon$ and $\rho$ is identifiability inducing.*

## 2.2 Multi-valued Functions

Throughout the proofs, we will make use of the notion of multi-valued functions and their continuity. A multi-valued function is a mapping $F : A \to \mathcal{P}(B)$ from a set $A$ to the power set $\mathcal{P}(B)$ of some set $B$. To define the continuity of $F$, we recall the Hausdorff distance [10, 16] between sets. Let $d_B$ be a distance function over a set $B$, the Hausdorff distance between two subsets $E_1, E_2$ of $B$ is defined as follows:

$$d_{\mathcal{H}}(E_1, E_2) := \max \left\{ \sup_{b_1 \in E_1} \inf_{b_2 \in E_2} d_B(b_1, b_2), \sup_{b_2 \in E_2} \inf_{b_1 \in E_1} d_B(b_1, b_2) \right\} \tag{3}$$

In general, the Hausdorff distance serves as an extended pseudo-metric, i.e., satisfies $d_{\mathcal{H}}(E, E) = 0$ for all $E$, is symmetric and satisfies the triangle inequality, however, it can attain infinite values and there might be $E_1 \neq E_2$, such that, $d_{\mathcal{H}}(E_1, E_2) = 0$. When considering the space $\mathcal{C}(B)$ of non-empty compact subsets of $B$, the Hausdorff distance becomes a metric.

**Definition 7** (Continuous multi-valued functions). *Let metric spaces $(A, d_A)$ and $(B, d_B)$ and multi-valued function $F : A \to \mathcal{C}(B)$. Then, we define:*

1. **Convergence:** *we denote $E = \lim_{a \to a_0} F(a)$, if $E$ is a compact subset of $B$ and it satisfies:*

$$\lim_{a \to a_0} d_{\mathcal{H}}(F(a), E) = 0 \tag{4}$$

2. **Continuity:** *we say that $F$ is continuous in $a_0$, if $\lim_{a \to a_0} F(a) = F(a_0)$.*

## 2.3 Lemmas

In this section, we provide several lemmas that will be useful throughout the proofs of the main results.

Let $[\mathbf{W}^1, \mathbf{b}^1]$ and $[\mathbf{W}^2, \mathbf{b}^2]$ be two parameterizations. We denote by $[\mathbf{W}^1, \mathbf{b}^1] - [\mathbf{W}^2, \mathbf{b}^2] = [\mathbf{W}^1 - \mathbf{W}^2, \mathbf{b}^1 - \mathbf{b}^2]$ the element-wise subtraction between the two parameterizations. In addition, we define the $L_2$-norm of $[\mathbf{W}, \mathbf{b}]$ to be:

$$\big\|[\mathbf{W}, \mathbf{b}]\big\|_2 := \|\text{vec}([\mathbf{W}, \mathbf{b}])\|_2 := \sqrt{\sum_{i=1}^{k} (\|W^i\|_2^2 + \|b^i\|_2^2)} \tag{5}$$

**Lemma 2.** *Let $f(x; [\mathbf{W}^1, \mathbf{b}^1])$ and $f(x; [\mathbf{W}^2, \mathbf{b}^2])$ be two neural networks. Then, for a given isomorphism $\pi$, we have:*

$$\pi \circ [\mathbf{W}^1, \mathbf{b}^1] - \pi \circ [\mathbf{W}^2, \mathbf{b}^2] = \pi \circ [\mathbf{W}^1 - \mathbf{W}^2, \mathbf{b}^1 - \mathbf{b}^2] \tag{6}$$

*and*

$$\left\| \pi \circ [\mathbf{W}, \mathbf{b}] \right\|_2 = \left\| [\mathbf{W}, \mathbf{b}] \right\|_2 \tag{7}$$

*Proof.* Follows immediately from the definition of isomorphisms.

**Lemma 3.** *Let $\sigma : \mathbb{R} \to \mathbb{R}$ be a $L$-Lipschitz continuous activation function, such that, $\sigma(0) = 0$. Let $f(\cdot; [\mathbf{W}, 0]) : \mathbb{R}^m \to \mathbb{R}$ be a neural network with zero biases. Then, for any $x \in \mathbb{R}^m$, we have:*

$$\|f(x; [\mathbf{W}, 0])\|_1 \leq L^{k-1} \cdot \|x\|_1 \prod_{i=1}^{k} \|W^i\|_1 \tag{8}$$

*Proof.* Let $z = W^{k-1} \cdot \sigma(\dots \sigma(W^1 x))$. We have:

$$\begin{aligned}
\|f(x; [\boldsymbol{W}, 0])\|_1 &\leq \|W^k \cdot \sigma(z)\|_1 \\
&\leq \|W^k \cdot \sigma(z)\|_1 \\
&= \|W^k\|_1 \cdot \|\sigma(z) - \sigma(0)\|_1 \\
&\leq \|W^k\|_1 \cdot L \cdot \|z\|_1
\end{aligned} \tag{9}$$

and by induction we have the desired. $\square$

**Lemma 4.** *Let $\sigma : \mathbb{R} \to \mathbb{R}$ be a $L$-Lipschitz continuous activation function, such that, $\sigma(0) = 0$. Let $f(\cdot; [\mathbf{W}, \mathbf{b}])$ be a neural network. Then, the Lipschitzness of $f(\cdot; [\mathbf{W}, \mathbf{b}])$ is given by:*

$$\mathrm{Lip}(f(\cdot; [\mathbf{W}, \mathbf{b}])) \leq L^{k-1} \cdot \prod_{i=1}^{k} \|W^i\|_1 \tag{10}$$

*Proof.* Let $z_i = W^{k-1} \cdot \sigma(\dots \sigma(W^1 x_i + b^1))$ for some $x_1$ and $x_2$. We have:

$$\begin{aligned}
\|f(x_1; [\boldsymbol{W}, \boldsymbol{b}]) - f(x_2; [\boldsymbol{W}, \boldsymbol{b}])\|_1 &\leq \|W^k \cdot \sigma(z_1) - W^k \cdot \sigma(z_2)\|_1 \\
&\leq \|W^k \cdot (\sigma(z_1 + b^{k-1}) - \sigma(z_2 + b^{k-1}))\|_1 \\
&= \|W^k\|_1 \cdot \|\sigma(z_1 + b^{k-1}) - \sigma(z_2 + b^{k-1})\|_1 \\
&\leq \|W^k\|_1 \cdot L \cdot \|z_1 - z_2\|_1
\end{aligned} \tag{11}$$

and by induction we have the desired. $\square$

Throughout the appendix, a function $y \in \mathbb{Y}$ is called *normal* with respect to $f$, if it has a best approximator $f \in f$, such that, $f \in f_n$.

**Lemma 5.** *Let $f$ be a class of neural networks. Let $y$ be a target function. Assume that $y$ has a best approximator $f \in f$. If $y \notin f$, then, $f \in f_n$.*

*Proof.* Let $f(\cdot; [\boldsymbol{W}, \boldsymbol{b}]) \in f$ be the best approximator of $y$. Assume it is not normal. Then, $f(\cdot; [\boldsymbol{W}, \boldsymbol{b}])$ has at least one zero neuron or at least one pair of clone neurons. Assume it has a zero neuron. Hence, by removing the specified neuron, we achieve a neural network of architecture smaller than $f$ that achieves the same approximation error as $f$ does. This is in contradiction to Assumption 2. For clone neurons, we can simply merge them into one neuron and obtain a smaller architecture that achieves the same approximation error, again, in contradiction to Assumption 2. $\square$

**Lemma 6.** *Let $f$ be a class of functions with a continuous activation function $\sigma$. Let $\mathbb{Y}$ be a class of target functions. Then, the function $\|f(\cdot; \theta) - y\|_\infty$ is continuous with respect to both $\theta$ and $y$ (simultaneously).*

*Proof.* Let sequences $\theta_n \to \theta_0$ and $y_n \to y_0$. By the reversed triangle inequality, we have:

$$\left| \|f(\cdot;\theta_n) - y_n\|_\infty - \|f(\cdot;\theta_0) - y_0\|_\infty \right| \leq \|f(\cdot;\theta_n) - f(\cdot;\theta_0)\|_\infty + \|y_n - y_0\|_\infty \tag{12}$$

Since $\theta_n \to \theta_0$ and $f$ is continuous with respect to $\theta$, we have: $\|f(\cdot;\theta_n) - f(\cdot;\theta_0)\|_\infty \to 0$. Hence, the upper bound tends to 0. $\qquad\square$

**Lemma 7.** *Let $f$ be a class of functions with a continuous activation function $\sigma$. Let $\mathbb{Y}$ be a closed class of target functions. Then, the function $F(y) := \min_{\theta \in \Theta_f} \|f(\cdot;\theta) - y\|_\infty$ is continuous with respect to $y$.*

*Proof.* Let $\{y_n\}_{n=1}^\infty \subset \mathbb{Y}$ be a sequence that converges to some $y_0 \in \mathbb{Y}$. Assume by contradiction that:

$$\lim_{n \to \infty} F(y_n) \neq F(y_0) \tag{13}$$

Then, there is a sub-sequence $y_{n_k}$ of $y_n$, such that, $\forall k \in \mathbb{N} : F(y_{n_k}) - F(y_0) > \Delta$ or $\forall k \in \mathbb{N} : F(y_0) - F(y_{n_k}) > \Delta$ for some $\Delta > 0$. Let $\theta_0$ be the minimizer of $\|f(\cdot;\theta) - y_0\|_\infty$. With no loss of generality, we can assume the first option. We notice that:

$$F(y_{n_k}) \leq \|f(\cdot;\theta_0) - y_{n_k}\|_\infty \leq \|f(\cdot;\theta_0) - y_0\|_\infty + \|y_{n_k} - y_0\|_\infty \leq F(y_0) + \delta_k \tag{14}$$

where $\delta_k := \|y_{n_k} - y_0\|_\infty$ tends to 0. This contradicts the assumption that $F(y_{n_k}) > F(y_0) + \Delta$. $\quad\square$

Throughout the appendix, we will make use of the following notation. Let $y \in \mathbb{Y}$ be a function and $f$ a class of functions, we define:

$$M[y; f] := \arg\min_{\theta \in \Theta_f} \|f(\cdot;\theta) - y\|_\infty \tag{15}$$

**Lemma 8.** *Let $f$ be a class of neural networks with a continuous activation function $\sigma$. Let $\mathbb{Y}$ be a class of target functions. Denote by $f_y$ the unique approximator of $y$ within $f$. Then, $f_y$ is continuous with respect to $y$.*

*Proof.* Let $y_0 \in \mathbb{Y}$ be some function. Assume by contradiction that there is a sequence $y_n \to y_0$, such that, $g_n := f_{y_n} \not\to f_{y_0}$. Then, $g_n$ has a sub-sequence that has no cluster points or it has a cluster point $h \neq f_{y_0}$.

**Case 1:** Let $g_{n_k}$ be a sub-sequence of $g_n$ that has no cluster points. By Assumption 1, there is a sequence $\theta_{n_k} \in \cup_{k=1}^\infty M[y_{n_k}; f]$ that is bounded in $\mathbb{B} = \{\theta \mid \|\theta\|_2 \leq B\}$. By the Bolzano-Weierstrass' theorem, it includes a convergent sub-sequence $\theta_{n_{k_i}} \to \theta_0$. Therefore, we have:

$$\|f(\cdot;\theta_{n_{k_i}}) - f(\cdot;\theta_0)\|_\infty \to 0 \tag{16}$$

Hence, $g_{n_k}$ has a cluster point $f(\cdot;\theta_0)$ in contradiction.

**Case 2:** Let sub-sequence $f_{y_{n_k}}$ that converge to a function $h \neq f_{y_0}$. We have:

$$\|h - y_0\|_\infty \leq \|f_{y_{n_k}} - h\|_\infty + \|f_{y_{n_k}} - y_{n_k}\|_\infty + \|y_{n_k} - y_0\|_\infty \tag{17}$$

By Lem. 7,

$$\|f_{y_{n_k}} - y_{n_k}\|_\infty \to \|f_{y_0} - y_0\|_\infty \tag{18}$$

and also $y_{n_k} \to y_0$, $f_{y_{n_k}} \to h$. Therefore, we have:

$$\|h - y_0\|_\infty \leq \|f_{y_0} - y_0\|_\infty \tag{19}$$

Hence, since $f_{y_0}$ is the unique minimizer, we conclude that $h = f_{y_0}$ in contradiction.

Therefore, we conclude that $f_{y_n}$ converges and by the analysis in Case 2 it converges to $f_{y_0}$. $\quad\square$

# 3 Proofs of the Main Results

## 3.1 Proving Assumption 2 for Shallow Networks

**Lemma 9.** *Let $\mathbb{Y} = C([-1,1]^m)$ be the class of continuous functions $y : [-1,1]^m \to \mathbb{R}$. Let $f$ be a class of 2-layered neural networks of width $d$ with $\sigma$ activations, where $\sigma$ is either $\tanh$ or sigmoid. Let $y \in \mathbb{Y}$ be some function to be approximated. Let $f'$ be a class of neural networks that is resulted by adding a neuron to the hidden layer of $f$. If $y \notin f$ then, $\inf_{\theta \in \Theta_f} \|f(\cdot;\theta) - y\|_2^2 > \inf_{\theta \in \Theta_{f'}} \|f(\cdot;\theta) - y\|_2^2$. The same holds for $\sigma = ReLU$ when $m = 1$.*

*Proof.* We divide the proof into two parts. In the first part we prove the claim for neural networks with ReLU activations and in the second part, for the $\tanh$ and sigmoid activations.

**ReLU activations**　Let $y \in \mathbb{Y}$ be a non-piecewise linear function. Let $f \in f$ be the best approximator of $y$. Since $f$ is a 2-layered neural network, it takes the form:

$$f(x) = \sum_{i=1}^{d} \beta_i \cdot \sigma(\alpha_i x + \gamma_i) \tag{20}$$

By [3], we note that $f$ is a piece-wise linear function with $k$ pieces. We denote the end-points of those pieces by: $-1 = c_0, \ldots, c_k = 1$. Since $y$ is a non-piecewise linear function, there exists a pair $c_i, c_{i+1}$, where $y$ is non-linear on $[c_i, c_{i+1}]$. With no loss of generality, we assume that $y$ is non-linear on the first segment. We note that $f$ equals some linear function $ax + b$ over the segment $[-1, c_1]$. We would like to prove that one is able to add a new neuron $n(x) = \beta_{d+1} \cdot \sigma(\gamma_{d+1} - x)$ to $f$, for some $-1 < \gamma_{d+1} < c_1$, such that, $f(x) + n(x)$ strictly improves the approximation of $f$. First, we notice that this neuron is non-zero only when $x < \gamma_{d+1}$. Therefore, for any $\beta_{d+1} \in \mathbb{R}$ and $-1 < \gamma_{d+1} < c_1$, $f(x) + n(x) = f(x) \in [c_1, 1]$. In particular, the approximation error of $f(x) + n(x)$ over $[c_1, 1]$ is the same as $f$'s. For simplicity, we denote $\gamma := \gamma_{d+1}$ and $\beta := \beta_{d+1}$. Assume by contradiction that there are no such $\gamma$ and $\beta$. Therefore, for each $\gamma \in [-1, c_1]$, $ax + b$ is the best linear approximator of $y(x)$ in the segment $[-1, \gamma]$. Hence, for each $\gamma \in [-1, c_1]$, $\beta = 0$ is the minimizer of $\int_{-1}^{\gamma} (y(x) - (\beta(\gamma - x) + ax + b))^2 \, dx$. In particular, we have:

$$\left. \frac{\int_{-1}^{\gamma} (y(x) - (\beta(\gamma - x) + ax + b))^2 \, dx}{\partial \beta} \right|_{\beta=0} = 0 \tag{21}$$

By differentiation under the integral sign:

$$
\begin{aligned}
Q(\beta, \gamma) =& \frac{\int_{-1}^{\gamma} (y(x) - (\beta(\gamma - x) + ax + b))^2 \, dx}{\partial \beta} \\
& \int_{-1}^{\gamma} \frac{(y(x) - (\beta(\gamma - x) + ax + b))^2}{\partial \beta} \, dx \\
=& \int_{-1}^{\gamma} 2(y(x) - (\beta(\gamma - x) + ax + b)) \cdot (x - \gamma) \, dx \\
=& 2 \int_{-1}^{\gamma} y(x)x \, dx - 2\gamma \int_{-1}^{\gamma} y(x) \, dx + 2 \int_{-1}^{\gamma} \beta(\gamma - x)^2 \, dx + 2 \int_{-1}^{\gamma} (ax + b)(\gamma - x) \, dx \\
=& 2 \int_{-1}^{\gamma} y(x)x \, dx - 2\gamma \int_{-1}^{\gamma} y(x) \, dx + p(\beta, \gamma)
\end{aligned}
\tag{22}
$$

where $p(\beta, \gamma)$ is a third degree polynomial with respect to $\gamma$. We denote by $Y(x)$ the primitive function of $y(x)$, and by $\mathcal{Y}(x)$ the primitive function of $Y(x)$. By applying integration by parts, we have:

$$\int_{-1}^{\gamma} y(x)x \, dx = Y(\gamma) \cdot \gamma - (\mathcal{Y}(\gamma) - \mathcal{Y}(-1)) \tag{23}$$

In particular,

$$
\begin{aligned}
Q(\beta, \gamma) =& 2\gamma(Y(\gamma) - Y(-1)) - 2(Y(\gamma) \cdot \gamma - \mathcal{Y}(\gamma) + \mathcal{Y}(-1)) + p(\beta, \gamma) \\
=& 2\gamma Y(\gamma) - 2\gamma Y(-1) - 2\gamma Y(\gamma) - 2\mathcal{Y}(\gamma) + 2\mathcal{Y}(-1) + p(\beta, \gamma) \\
=& -2\mathcal{Y}(\gamma) + [-2\gamma Y(-1) + 2\mathcal{Y}(-1) + p(\beta, \gamma)]
\end{aligned}
\tag{24}
$$

We note that the function $q(\beta, \gamma) := -2\gamma Y(-1) + 2\mathcal{Y}(-1) + p(\beta, \gamma)$ is a third degree polynomial with respect to $\gamma$ (for any fixed $\beta$). In addition, by Eq. 21, we have, $Q(0, \gamma) = 0$ for any value of $\gamma \in (-1, c_1)$. Hence, $\mathcal{Y}$ is a third degree polynomial over $[-1, c_1]$. In particular, $y$ is a linear function over $[-1, c_1]$, in contradiction. Therefore, there exist values $\gamma \in (-1, c_1)$ and $\beta \in \mathbb{R}$, such that, $f(x) + n(x)$ strictly improves the approximation of $f$.

**Sigmoidal activations**    Let $y \in \mathbb{Y}$ be a target function that is not a member of $f$. Let $f \in f$ be the best approximator of $y$. In particular, $f \neq y$. Since $f$ is a 2-layered neural network, it takes the form:

$$f(x) = \sum_{i=1}^{d} \beta_i \cdot \sigma(\langle \alpha_i, x \rangle + \gamma_i) \tag{25}$$

where $\sigma : \mathbb{R} \to \mathbb{R}$ is either $\tanh$ or the sigmoid activation function, $\beta_i, \gamma_i \in \mathbb{R}$ and $\alpha_i \in \mathbb{R}^m$.

We would like to show the existence of a neuron $n(x) = \beta \cdot \sigma(\langle a, x \rangle + b)$, such that, $f + n$ has a smaller approximation error with respect to $y$, compared to $f$. Assume the contrary by contradiction. Then, for any $a \in \mathbb{R}^m, b \in \mathbb{R}$, we have:

$$\frac{\int_{[-1,1]^m} (y(x) - (\beta \cdot \sigma(\langle a, x \rangle + b) + f(x)))^2 \, dx}{\partial \beta} \Bigg|_{\beta=0} = 0 \tag{26}$$

We denote by $q(x) := y(x) - f(x)$. By differentiating under the integral sign:

$$\begin{aligned} Q(\beta, a, b) := & \frac{\int_{[-1,1]^m} (y(x) - (\beta \cdot \sigma(\langle a, x \rangle + b) + f(x)))^2 \, dx}{\partial \beta} \\ = & -2 \int_{[-1,1]^m} \beta \cdot \sigma(\langle a, x \rangle + b)^2 \, dx + 2 \int_{[-1,1]^m} q(x) \cdot \sigma(\langle a, x \rangle + b) \, dx \end{aligned} \tag{27}$$

Therefore, since $Q(\beta, a, b) = 0$, we have:

$$\beta = \frac{\int_{[-1,1]^m} q(x) \cdot \sigma(\langle a, x \rangle + b) \, dx}{\int_{[-1,1]^m} \sigma(\langle a, x \rangle + b)^2 \, dx} \tag{28}$$

Since $\sigma$ is increasing, it is non-zero on any interval, and therefore, the denominator in Eq. 28 is strictly positive for all $a \in \mathbb{R}^m \setminus \{0\}, b \in \mathbb{R}$ and $a = 0, b \in \mathbb{R}$, such that, $\sigma(b) \neq 0$. In particular, for all such $a, b$, we have:

$$\int_{[-1,1]^m} q(x) \cdot \sigma(\langle a, x \rangle + b) \, dx = 0 \tag{29}$$

By the universal approximation theorem [5, 12], there exist $\hat{\beta}_j, \hat{b}_j \in \mathbb{R}$ and $\hat{a}_j \in \mathbb{R}^m$, such that,

$$f(x) - y(x) = \sum_{j=1}^{\infty} \hat{\beta}_j \cdot \sigma(\langle \hat{a}_j, x \rangle + \hat{b}_j) \tag{30}$$

where $\hat{a}_j \in \mathbb{R}^m \setminus \{0\}, \hat{b}_j \in \mathbb{R}$ and $\hat{a}_j = 0, \hat{b}_j \in \mathbb{R}$, such that, $\sigma(\hat{b}_j) \neq 0$. The convergence of the series is uniform over $[-1, 1]^m$. In particular, the series $q(x) \cdot \sum_{j=1}^{k} \hat{\beta}_j \cdot \sigma(\langle \hat{a}_j, x \rangle + \hat{b}_j)$ converge uniformly as $k \to \infty$. Therefore, by Eq. 29 and the linearity of integration, we have:

$$\int_{[-1,1]^m} q(x) \cdot \sum_{j=1}^{\infty} \hat{\beta}_j \cdot \sigma(\langle \hat{a}_j, x \rangle + \hat{b}_j) \, dx = 0 \tag{31}$$

This implies that $\int_{[-1,1]^m} q(x)^2 \, dx = 0$. Since $q$ is a continuous function, it must be the zero function to satisfy this condition. Differently put, $f = y$ in contradiction. $\qquad \square$

## 3.2    Existence of a continuous selector

In this section, we prove that for any compact set $\mathbb{Y}' \subset \mathbb{Y}$, if any $y \in \mathbb{Y}'$ cannot be represented as a neural network with $\sigma$ activations, then, there exists a continuous selector $S : \mathbb{Y}' \to \mathbb{R}^{N_f}$ that returns the parameters of a good approximator $f(\cdot; S(y))$ of $y$. Before we provide a formal statement of the proof, we give an informal overview of the main arguments.

**Proof sketch of Lem. 16** Let $\mathbb{Y}' \subset \mathbb{Y}$ be a compact class of target functions, such that, any $y \in \mathbb{Y}'$ cannot be represented as a neural network with $\sigma$ activations. We recall that, by Lem. 1, one can approximate $\sigma$ using a continuous, identifiability inducing, activation function $\rho : \mathbb{R} \to \mathbb{R}$, up to any error $\epsilon > 0$ of our choice. By Assumption 1, for each $y \in \mathbb{Y}$, there exists a unique best function approximator $g(\cdot; \theta_y) \in g$ of $y$. Here, $g$ is the class of neural networks of the same architecture as $f$ except the activations are $\rho$. By Def. 6, $\theta_y$ is unique up to isomorphisms, assuming that $g(\cdot; \theta_y)$ is normal (see Def. 5).

In Lem. 11 we show that for any compact set $\mathbb{Y}' \subset \mathbb{Y}$, if $g(\cdot; \theta_y)$ is normal for all $y \in \mathbb{Y}'$, then, there exists a continuous selector $S : \mathbb{Y}' \to \mathbb{R}^{N_g}$ that returns the parameters of a best approximator $g(\cdot; S(y))$ of $y$. Therefore, in order to show the existence of $S$, we need to prove that $g(\cdot; \theta_y)$ is normal for all $y \in \mathbb{Y}'$.

Since any function $y \in \mathbb{Y}'$ cannot be represented as a neural network with $\sigma$ activations, $\inf_{y \in \mathbb{Y}'} \inf_{\theta_f} \|f(\cdot; \theta) - y\|_\infty$ is strictly larger than zero (see Lem. 12). In particular, by taking $\rho$ to be close enough to $\sigma$, we can ensure that, $\inf_{y \in \mathbb{Y}'} \inf_{\theta_f} \|g(\cdot; \theta) - y\|_\infty$ is also strictly larger than zero. This, together with Assumption 2, imply that $g(\cdot; \theta_y)$ is normal for all $y \in \mathbb{Y}'$ (see Lem. 5). Hence, there exists a continuous selector $S$ for $\mathbb{Y}'$ with respect to the class $g$. Finally, using Lem. 14, one can show that if $\rho$ is close enough to $\sigma$, $S$ is a good parameter selector for $f$ as well.

**Lemma 10.** *Let $\rho : \mathbb{R} \to \mathbb{R}$ be a continuous, identifiability inducing, activation function. Let $f$ be a class of neural networks with $\rho$ activations and $\Theta_f = \mathbb{B}$ be the closed ball in the proof of Lem. 8. Let $\mathbb{Y}$ be a class of normal target functions with respect to $f$. Then, $M[y; f] := \arg\min_{\theta \in \mathbb{B}} \|f(\cdot; \theta) - y\|_\infty$ is a continuous multi-valued function of $y$.*

*Proof.* Assume by contradiction that $M$ is not continuous. We distinguish between two cases:

1. There exists a sequence $y_n \to y$ and constant $c > 0$, such that,

$$\sup_{\theta \in M[y; f]} \inf_{\theta \in M[y_n; f]} \|\theta_1 - \theta_2\|_2 > c > 0 \tag{32}$$

2. There exists a sequence $y_n \to y$ and constant $c > 0$, such that,

$$\sup_{\theta_1 \in M[y_n; f]} \inf_{\theta_2 \in M[y; f]} \|\theta_1 - \theta_2\|_2 > c > 0 \tag{33}$$

**Case 1:** We denote by $\theta_1$ a member of $M[y; f]$ that satisfies:

$$\forall n \in \mathbb{N} : \inf_{\theta_2 \in M[y_n; f]} \|\theta_1 - \theta_2\|_2 > c > 0 \tag{34}$$

The set $\cup_{n=1}^\infty M[y_n; f] \subset \Theta_f$ is a bounded subset of $\mathbb{R}^N$, and therefore by the Bolzano-Weierstrass theorem, for any sequence $\{\theta_2^n\}_{n=1}^\infty$, such that, $\theta_2^n \in M[y_n; f]$, there is a sub-sequence $\{\theta_2^{n_k}\}_{k=1}^\infty$ that converges to some $\theta_2^*$. We notice that:

$$\|f(\cdot; \theta_2^{n_k}) - y_{n_k}\|_\infty = \min_{\theta \in \Theta_f} \|f(\cdot; \theta) - y_{n_k}\|_\infty = F(y_{n_k}) \tag{35}$$

In addition, by the continuity of $F$, we have: $\lim_{k \to \infty} F(y_{n_k}) = F(y)$. By Lem. 6, we have:

$$\|f(\cdot; \theta_2^*) - y\|_\infty = F(y) \tag{36}$$

This yields that $\theta_2^*$ is a member of $M[y; f]$. Since $f_y := \arg\min_{f \in f} \|f - y\|_\infty$ is unique and normal, by the identifiability hypothesis, there is a function $\pi \in \Pi$, such that, $\pi(\theta_2^*) = \theta_1$. Since the function $\pi$ is continuous

$$\lim_{k \to \infty} \|\pi(\theta_2^{n_k}) - \theta_1\|_2 = \lim_{k \to \infty} \|\pi(\theta_2^{n_k}) - \pi(\theta_2^*)\|_2 = 0 \tag{37}$$

We notice that $\pi(\theta_2^{n_k}) \in M[y_{n_k}; f]$. Therefore, we have:

$$\lim_{k \to \infty} \inf_{\theta_2 \in M[y_{n_k}; f]} \|\theta_1 - \theta_2\| = 0 \tag{38}$$

in contradiction to Eq. 34.

**Case 2:** Let $\theta_1^n \in M[y_n; \mathfrak{f}]$ be a sequence, such that,

$$\inf_{\theta_2 \in M[y; \mathfrak{f}]} \|\theta_1^n - \theta_2\|_\infty > c \tag{39}$$

The set $\cup_{n=1}^\infty M[y_n; \mathfrak{f}] \subset \Theta_\mathfrak{f}$ is a bounded subset of $\mathbb{R}^N$, and therefore by the Bolzano-Weierstrass theorem, there is a sub-sequence $\theta_1^{n_k}$ that converges to some vector $\theta_0$. The function $\|f(\cdot; \theta) - y\|_\infty$ is continuous with respect to $\theta$ and $y$. Therefore,

$$\lim_{k\to\infty} \min_{\theta \in \Theta_\mathfrak{f}} \|f(\cdot; \theta) - y_{n_k}\|_\infty = \lim_{k\to\infty} \|f(\cdot; \theta_1^{n_k}) - y_{n_k}\|_\infty = \|f(\cdot; \theta_0) - y\|_\infty \tag{40}$$

By Lem. 7, $\|f(\cdot; \theta_0) - y\|_\infty = \min_{\theta \in \Theta_\mathfrak{f}} \|f(\cdot; \theta) - y\|_\infty$. In particular, $\theta_0 \in M[y; \mathfrak{f}]$, in contradiction to Eq. 39. $\qquad\square$

**Lemma 11.** *Let $\rho : \mathbb{R} \to \mathbb{R}$ be a continuous, identifiability inducing, activation function. Let $\mathfrak{f}$ be a class of neural networks with $\rho$ activations and $\Theta_\mathfrak{f} = \mathbb{B}$ be the closed ball in the proof of Lem. 8. Let $\mathbb{Y}$ be a compact class of normal target functions with respect to $\mathfrak{f}$. Then, there is a continuous selector $S : \mathbb{Y} \to \Theta_\mathfrak{f}$, such that, $S(y) \in M[y; \mathfrak{f}]$.*

*Proof.* Let $y_0$ be a member of $\mathbb{Y}$. We notice that $M[y_0; \mathfrak{f}]$ is a finite set. We denote its members by: $M[y_0; \mathfrak{f}] = \{\theta_1^0, \ldots, \theta_k^0\}$. Then, we claim that there is a small enough $\epsilon := \epsilon(y_0) > 0$ (depending on $y_0$), such that, $S$ that satisfies $S(y_0) = \theta_1^0$ and $S(y) = \arg\min_{\theta \in M[y; \mathfrak{f}]} \|\theta - \theta_0\|_2$ for all $y \in \mathbb{B}_\epsilon(y_0)$, is continuous in $\mathbb{B}_\epsilon(y_0)$. The set $\mathbb{B}_\epsilon(y_0) := \{y \mid \|y - y_0\|_\infty < \epsilon\}$ is the open ball of radius $\epsilon$ around $y_0$. We denote

$$c := \min_{\pi_1 \neq \pi_2 \in \Pi} \|\pi_1 \circ S(y_0) - \pi_2 \circ S(y_0)\|_2 > 0 \tag{41}$$

This constant exists since $\Pi$ is a finite set of transformations and $\mathbb{Y}$ is a class of normal functions. In addition, we select $\epsilon$ to be small enough to suffice that:

$$\max_{y \in \mathbb{B}_\epsilon(y_0)} \|S(y) - S(y_0)\|_2 < c/4 \tag{42}$$

Assume by contradiction that there is no such $\epsilon$. Then, for each $\epsilon_n = 1/n$ there is a function $y_n \in \mathbb{B}_{\epsilon_n}(y_0)$, such that,

$$\|S(y) - S(y_0)\|_2 \geq c/4 \tag{43}$$

Therefore, we found a sequence $y_n \to y_0$ that satisfies:

$$M[y_n; \mathfrak{f}] \not\to M[y_0; \mathfrak{f}] \tag{44}$$

in contradiction to the continuity of $M$.

For any given $y_1, y_2 \in \mathbb{B}_\epsilon(y_0)$ and $\pi_1 \neq \pi_2 \in \Pi$, by the triangle inequality, we have:

$$\begin{aligned}
\|\pi_1 \circ S(y_1) - \pi_2 \circ S(y_2)\|_2 \geq &\|\pi_1 \circ S(y_0) - \pi_2 \circ S(y_2)\|_2 - \|\pi_1 \circ S(y_1) - \pi_1 \circ S(y_0)\|_2 \\
\geq &\|\pi_1 \circ S(y_0) - \pi_2 \circ S(y_0)\|_2 - \|\pi_1 \circ S(y_1) - \pi_1 \circ S(y_0)\|_2 \\
&- \|\pi_2 \circ S(y_0) - \pi_2 \circ S(y_2)\|_2 \\
= &\|\pi_1 \circ S(y_0) - \pi_2 \circ S(y_0)\|_2 - \|S(y_1) - S(y_0)\|_2 \\
&- \|S(y_0) - S(y_2)\|_2 \\
\geq &c - 2c/4 > c/2
\end{aligned} \tag{45}$$

In particular, $\|\pi \circ S(y_1) - S(y_2)\|_2 > c/2$ for every $\pi \neq \text{Id}$.

Since $M$ is continuous, for any sequence $y_n \to y \in \mathbb{B}_\epsilon(y_0)$, there are $\pi_n \in \Pi$, such that:

$$\lim_{n\to\infty} \pi_n \circ S(y_n) = S(y) \tag{46}$$

Therefore, by the above inequality, we address that for any large enough $n$, $\pi_n = \text{Id}$. In particular, for any sequence $y_n \to y$, we have:

$$\lim_{n\to\infty} S(y_n) = S(y) \tag{47}$$

This implies that $S$ is continuous in any $y \in \mathbb{B}_\epsilon(y_0)$.

We note that $\{\mathbb{B}_{\epsilon(y_0)}(y_0)\}_{y_0 \in \mathbb{Y}}$ is an open cover of $\mathbb{Y}$. In particular, since $\mathbb{Y}$ is compact, there is a finite sub-cover $\{C_i\}_{i=1}^T$ of $\mathbb{Y}$. In addition, we denote by $\{c_i\}_{i=1}^T$ the corresponding constants

in Eq. 41. Next, we construct the continuous function $S$ inductively. We denote by $S_i$ the locally continuous function that corresponds to $C_i$. For a given pair of sets $C_{i_1}$ and $C_{i_2}$ that intersect, we would like to construct a continuous function over $C_{i_1} \cup C_{i_2}$. First, we would like to show that there is an isomorphism $\pi$, such that, $\pi \circ S_{i_2}(y) = S_{i_1}(y)$ for all $y \in C_{i_1} \cap C_{i_2}$. Assume by contradiction that there is no such $\pi$. Then, let $y_1 \in C_{i_1} \cap C_{i_2}$ and $\pi_1$, such that, $\pi_1 \circ S_{i_2}(y_1) = S_{i_1}(y_1)$. We denote by $y_2 \in C_{i_1} \cap C_{i_2}$ a member, such that, $\pi_1 \circ S_{i_2}(y_2) \neq S_{i_1}(y_2)$. Therefore, we take a isomorphism $\pi_2 \neq \pi_1$, that satisfies $\pi_2 \circ S_{i_2}(y_2) = S_{i_1}(y_2)$. We note that:

$$\|\pi_1 \circ S_{i_2}(y_1) - \pi_2 \circ S_{i_2}(y_2)\|_2 > \max\{c_{i_1}, c_{i_2}\}/2 \tag{48}$$

on the other hand:

$$\|\pi_1 \circ S_{i_2}(y_1) - \pi_2 \circ S_{i_2}(y_2)\|_2 = \|S_{i_1}(y_1) - S_{i_1}(y_2)\|_2 < c_{i_1}/4 \tag{49}$$

in contradiction.

Hence, let $\pi$ be such isomorphism. To construct a continuous function over $C_{i_1} \cup C_{i_2}$ we proceed as follows. First, we replace $S_{i_2}$ with $\pi \circ S_{i_2}$ and define a selection function $S_{i_1, i_2}$ over $C_{i_1} \cup C_{i_2}$ to be:

$$S_{i_1, i_2}(y) := \begin{cases} S_{i_1}(y) & \text{if } , y \in C_{i_1} \\ \pi \circ S_{i_2}(y) & \text{if } , y \in C_{i_2} \end{cases} \tag{50}$$

Since each one of the functions $S_{i_1}$ and $\pi \circ S_{i_2}$ are continuous, they conform on $C_{i_1} \cap C_{i_2}$ and the sets $C_{i_1}$ and $C_{i_2}$ are open, $S_{i_1, i_2}$ is continuous over $C_{i_1} \cup C_{i_2}$. We define a new cover $(\{C_i\}_{i=1}^T \setminus \{C_{i_1}, C_{i_2}\}) \cup \{C_{i_1} \cup C_{i_2}\}$ of size $T - 1$ with locally continuous selection functions $S'_1, \ldots, S'_{T-1}$. By induction, we can construct $S$ over $\mathbb{Y}$. $\qquad\square$

**Lemma 12.** *Let $f$ be a class of neural networks with a continuous activation function $\sigma$. Let $\mathbb{Y}$ be a compact class of target functions. Assume that any $y \in \mathbb{Y}$ cannot be represented as a neural network with $\sigma$ activations. Then,*

$$\inf_{y \in \mathbb{Y}} \inf_{\theta \in \Theta_f} \|f(\cdot; \theta) - y\|_\infty > c_2 \tag{51}$$

*for some constant $c_2 > 0$.*

*Proof.* Assume by contradiction that:

$$\inf_{y \in \mathbb{Y}} \inf_{\theta \in \Theta_f} \|f(\cdot; \theta) - y\|_\infty = 0 \tag{52}$$

Then, there is a sequence $y_n \in \mathbb{Y}$, such that:

$$\inf_{\theta \in \Theta_f} \|f(\cdot; \theta) - y_n\|_\infty \to 0 \tag{53}$$

Since $\mathbb{Y}$ is compact, there exists a converging sub-sequence $y_{n_k} \to y_0 \in \mathbb{Y}$. By Lem. 7, we have:

$$\inf_{\theta \in \Theta_f} \|f(\cdot; \theta) - y_0\|_\infty = 0 \tag{54}$$

This is in contradiction to the assumption that any $y \in \mathbb{Y}$ cannot be represented as a neural network with $\sigma$ activations. $\qquad\square$

**Lemma 13.** *Let $f$ be a class of neural networks with a continuous activation function $\sigma$. Let $\mathbb{Y}$ be a compact class of target functions. Assume that any $y \in \mathbb{Y}$ cannot be represented as a neural network with $\sigma$ activations. Then, there exists a closed ball $\mathbb{B}$ around $0$ in the Euclidean space $\mathbb{R}^{N_f}$, such that:*

$$\min_{\theta \in \mathbb{B}} \|f(\cdot; \theta) - y\|_\infty \leq 2 \inf_{\theta \in \Theta_f} \|f(\cdot; \theta) - y\|_\infty \tag{55}$$

*Proof.* Let $c_2 > 0$ be the constant from Lem. 12. By Lem. 12 and Lem. 7, $f_y$ is continuous over the compact set $\mathbb{Y}$. Therefore, there is a small enough $\delta > 0$, such that, for any $y_1, y_2 \in \mathbb{Y}$, such that, $\|y_1 - y_2\|_\infty < \delta$, we have: $\|f_{y_1} - f_{y_2}\|_\infty < c_2/2$. For each $y \in \mathbb{Y}$ we define $B(y) := \{y' \mid \|y - y'\|_\infty < \min\{c_2/2, \delta\}\}$. The sets $\{B(y)\}_{y \in \mathbb{Y}}$ form an open cover to $\mathbb{Y}$. Since $\mathbb{Y}$ is a compact set, it has a finite sub-cover $\{B(y_1), \ldots, B(y_k)\}$. For each $y' \in B(y_i)$, we have:

$$\begin{aligned} \|f_{y_i} - y'\|_\infty &\leq \|f_{y_i} - f_{y'}\|_\infty + \|f_{y'} - y'\|_\infty \\ &\leq c_2/2 + \|f_{y'} - y'\|_\infty \\ &\leq 2\|f_{y'} - y'\|_\infty \end{aligned} \tag{56}$$

Therefore, if we take $H = \{\theta_i\}_{i=1}^k$ for $\theta_i$, such that, $f(\cdot; \theta_i) = f_{y_i}$, we have:

$$\min_{i \in [n]} \|f(\cdot; \theta_i) - y\|_\infty \leq 2 \inf_{\theta \in \Theta_f} \|f(\cdot; \theta) - y\|_\infty \tag{57}$$

In particular, if we take $\mathbb{B}$ to be the closed ball around 0 that contains $H$, we have the desired. $\quad\square$

**Lemma 14.** *Let $\sigma : \mathbb{R} \to \mathbb{R}$ be a L-Lipschitz continuous activation function. Let $f$ be a class of neural networks with $\sigma$ activations. Let $\mathbb{Y}$ be a compact class of normal target functions with respect to $f$. Let $\rho$ be an activation function, such that, $\|\sigma - \rho\|_\infty < \delta$. Let $\mathbb{B} = \mathbb{B}_1 \cup \mathbb{B}_2$ be the closed ball around 0, where $\mathbb{B}_1$ is be the closed ball in the proof of Lem. 8 and $\mathbb{B}_2$ is the ball from Lem. 13. In addition, let $g$ be the class of neural networks of the same architecture as $f$ except the activations are $\rho$. Then, for any $\theta \in \mathbb{B}$, we have:*

$$\|f(\cdot; \theta) - g(\cdot; \theta)\|_\infty \leq c_1 \cdot \delta \tag{58}$$

*for some constant $c_1 > 0$ independent of $\delta$.*

*Proof.* We prove by induction that for any input $x \in \mathcal{X}$ the outputs the $i$'th layer of $f(\cdot; \theta)$ and $g(\cdot; \theta)$ are $\mathcal{O}(\delta)$-close to each other.

**Base case:** we note that:

$$\|\sigma(W^1 \cdot x + b^1) - \rho(W^1 \cdot x + b^1)\|_1 \leq \sum_{i=1}^{h_2} \left| \sigma(\langle W_i^1, x \rangle + b_i^1) - \rho(\langle W_i^1, x \rangle + b_i^1) \right|$$
$$\leq h_2 \cdot \delta =: c^1 \cdot \delta \tag{59}$$

Hence, the first layer's activations are $\mathcal{O}(\delta)$-close to each other.

**Induction step:** assume that for any two vectors of activations $x_1$ and $x_2$ in the $i$'th layer of the neural networks, we have:

$$\|x_1 - x_2\|_1 \leq c^i \cdot \delta \tag{60}$$

By the triangle inequality:

$$\begin{aligned}
&\|\sigma(W^{i+1} \cdot x_1 + b^{i+1}) - \rho(W^{i+1} x_2 + b^{i+1})\|_1 \\
\leq& \|\sigma(W^{i+1} \cdot x_1 + b^{i+1}) - \sigma(W^{i+1} x_2 + b^{i+1})\|_1 \\
&+ \|\sigma(W^{i+1} x_2 + b^{i+1}) - \rho(W^{i+1} x_2 + b^{i+1})\|_1 \\
\leq& L \cdot \|(W^{i+1} \cdot x_1 + b^{i+1}) - (W^{i+1} x_2 + b^{i+1})\|_1 \\
&+ \sum_{j=1}^{h_{i+2}} |\sigma(\langle W_j^{i+1}, x \rangle + b_j^{i+1}) - \rho(\langle W_j^{i+1}, x \rangle + b_j^{i+1})| \\
=& L \cdot \|W^{i+1}(x_1 - x_2)\|_1 + h_{i+2} \cdot \delta \\
\leq& L \cdot \|W^{i+1}\|_1 \cdot \|x_1 - x_2\|_1 + h_{i+2} \cdot \delta \\
\leq& L \cdot \|W^{i+1}\|_1 \cdot c^i \cdot \delta + h_{i+2} \cdot \delta \\
\leq& (h_{i+2} + L \cdot \|W^{i+1}\|_1 \cdot c^i) \cdot \delta
\end{aligned} \tag{61}$$

Since $\theta \in \mathbb{B}$ is bounded, each $\|W^{i+1}\|_1$ is bounded (for all $i \leq k$ and $\theta$). Hence, Eq. 58 holds for some constant $c_1 > 0$ independent of $\delta$. $\quad\square$

**Lemma 15.** *Let $\sigma : \mathbb{R} \to \mathbb{R}$ be a L-Lipschitz continuous activation function. Let $f$ be a class of neural networks with $\sigma$ activations. Let $\mathbb{Y}$ be a compact class of target functions. Assume that any $y \in \mathbb{Y}$ cannot be represented as a neural network with $\sigma$ activations. Let $\rho$ be an activation function, such that, $\|\sigma - \rho\|_\infty < \delta$. Let $\mathbb{B}$ be the closed ball from Lem. 14. In addition, let $g$ be the class of neural networks of the same architecture as $f$ except the activations are $\rho$. Then, for any $y \in \mathbb{Y}$, we have:*

$$\left| \min_{\theta \in \mathbb{B}} \|f(\cdot; \theta) - y\|_\infty - \min_{\theta \in \mathbb{B}} \|g(\cdot; \theta) - y\|_\infty \right| \leq c_1 \cdot \delta \tag{62}$$

*for $c_1$ from Lem. 14.*

*Proof.* By Lem. 14, for all $\theta \in \mathbb{B}$, we have:

$$\|f(\cdot;\theta) - y\|_\infty \leq \|g(\cdot;\theta) - y\|_\infty + c_1 \cdot \delta \tag{63}$$

In particular,

$$\min_{\theta \in \mathbb{B}} \|f(\cdot;\theta) - y\|_\infty \leq \min_{\theta \in \mathbb{B}} \|g(\cdot;\theta) - y\|_\infty + c_1 \cdot \delta \tag{64}$$

By a similar argument, we also have:

$$\min_{\theta \in \mathbb{B}} \|g(\cdot;\theta) - y\|_\infty \leq \min_{\theta \in \mathbb{B}} \|f(\cdot;\theta) - y\|_\infty + c_1 \cdot \delta \tag{65}$$

Hence,

$$\left| \min_{\theta \in \mathbb{B}} \|f(\cdot;\theta) - y\|_\infty - \min_{\theta \in \mathbb{B}} \|g(\cdot;\theta) - y\|_\infty \right| \leq c_1 \cdot \delta \tag{66}$$

$\square$

**Lemma 16.** *Let $\sigma : \mathbb{R} \to \mathbb{R}$ be a L-Lipschitz continuous activation function. Let $f$ be a class of neural networks with $\sigma$ activations. Let $\mathbb{Y}$ be a compact set of target functions. Assume that any $y \in \mathbb{Y}$ cannot be represented as a neural network with $\sigma$ activations. Then, for every $\hat{\epsilon} > 0$ there is a continuous selector $S : \mathbb{Y} \to \Theta_f$, such that, for all $y \in \mathbb{Y}$, we have:*

$$\|f(\cdot;S(y)) - y\|_\infty \leq 2 \inf_{\theta \in \Theta_f} \|f(\cdot;\theta) - y\|_\infty + \hat{\epsilon} \tag{67}$$

*Proof.* By Lem. 1, there exists a meromorphic function $\rho : D \to \mathbb{C}$, $\mathbb{R} \subset D$, $\rho(\mathbb{R}) \subset \mathbb{R}$ that is an identifiability inducing function, such that, $\|\sigma - \rho\|_\infty < \frac{1}{2c_2} \min(\hat{\epsilon}, c_1) =: \delta$, where $c_1$ and $c_2$ are the constants in Lems. 14 and 12. Since $\rho(\mathbb{R}) \subset \mathbb{R}$, and it is a meromorphic over $D$, it is continuous over $\mathbb{R}$ (the poles of $\rho$ are not in $\mathbb{R}$). We note that by Lems. 14 and 15, for any $y \in \mathbb{Y}$, we have:

$$\min_{\theta \in \mathbb{B}} \|g(\cdot;\theta) - y\|_\infty > c_2 - c_1 \cdot \delta > 0 \tag{68}$$

where $\mathbb{B}$ is the ball from Lem. 14. Therefore, by Lem. 5, each $y \in \mathbb{Y}$ is normal with respect to the class $g$. Hence, by Thm. 11, there is a continuous selector $S : \mathbb{Y} \to \mathbb{B}$, such that,

$$\|g(\cdot;S(y)) - y\|_\infty = \min_{\theta \in \mathbb{B}} \|g(\cdot;\theta) - y\|_\infty \tag{69}$$

By Lem. 15, we have:

$$\left| \min_{\theta \in \mathbb{B}} \|f(\cdot;\theta) - y\|_\infty - \|g(\cdot;S(y)) - y\|_\infty \right| \leq c_1 \cdot \delta \tag{70}$$

By the triangle inequality:

$$\begin{aligned} &\left| \min_{\theta \in \mathbb{B}} \|f(\cdot;\theta) - y\|_\infty - \|f(\cdot;S(y)) - y\|_\infty \right| \\ \leq &\left| \|f(\cdot;S(y)) - y\|_\infty - \|g(\cdot;S(y)) - y\|_\infty \right| + \left| \min_{\theta \in \mathbb{B}} \|f(\cdot;\theta) - y\|_\infty - \|g(\cdot;S(y)) - y\|_\infty \right| \end{aligned} \tag{71}$$

By Eq. 70 and Lem. 14, we have:

$$\left| \min_{\theta \in \mathbb{B}} \|f(\cdot;\theta) - y\|_\infty - \|f(\cdot;S(y)) - y\|_\infty \right| \leq 2c_1 \cdot \delta \tag{72}$$

Since $\delta < \hat{\epsilon}/2c_2$, we obtain the desired inequality:

$$\begin{aligned} \|f(\cdot;S(y)) - y\|_\infty &\leq \min_{\theta \in \mathbb{B}} \|f(\cdot;\theta) - y\|_\infty + \hat{\epsilon} \\ &\leq 2 \min_{\theta \in \Theta_f} \|f(\cdot;\theta) - y\|_\infty + \hat{\epsilon} \end{aligned} \tag{73}$$

$\square$

## 3.3 Proof of Thm. 1

Before we provide a formal statement of the proof, we introduce an informal outline of it.

**Proof sketch of Thm. 1**   In Lem. 16 we showed that for a compact class $\mathbb{Y}$ of target functions that cannot be represented as neural networks with $\sigma$ activations, there is a continuous selector $S(y)$ of parameters, such that,

$$\|f(\cdot; S(y)) - y\|_\infty \leq 3 \inf_{\theta \in \Theta_f} \|f(\cdot; \theta) - y\|_\infty \qquad (74)$$

Therefore, in this case, we have: $d_N(f; \mathbb{Y}) = \Theta(\tilde{d}_N(f; \mathbb{Y}))$. As a next step, we would like to apply this claim on $\mathbb{Y} := \mathcal{W}_{r,m}$ and apply the lower bound of $\tilde{d}_N(f; \mathcal{W}_{r,m}) = \Omega(N^{-r/m})$ to lower bound $d_N(f; \mathbb{Y})$. However, both of the classes $f$ and $\mathbb{Y}$ include constant functions, and therefore, we have: $f \cap \mathbb{Y} \neq \emptyset$. Hence, we are unable to assume that any $y \in \mathbb{Y}$ cannot be represented as a neural network with $\sigma$ activations.

To solve this issue, we consider a "wide" compact subset $\mathbb{Y}' = \mathcal{W}_{r,m}^\gamma$ of $\mathcal{W}_{r,m}$ that does not include any constant functions, but still satisfies $\tilde{d}_N(f; \mathcal{W}_{r,m}^\gamma) = \Omega(N^{-r/m})$. Then, assuming that any non-constant function $y \in \mathcal{W}_{r,m}$ cannot be represented as a neural network with $\sigma$ activations, implies that any $y \in \mathcal{W}_{r,m}^\gamma$ cannot be represented as a neural network with $\sigma$ activations. In particular, by Lem. 16, we obtain the desired lower bound: $d_N(f; \mathcal{W}_{r,m}) \geq d_N(f; \mathcal{W}_{r,m}^\gamma) = \Theta(\tilde{d}_N(f; \mathcal{W}_{r,m}^\gamma)) = \Omega(N^{-r/m})$.

For this purpose, we provide some technical notations. For a given function $f : [-1, 1]^m \to \mathbb{R}$, we denote:

$$\|h\|_r^{s,*} := \sum_{1 \leq |\mathbf{k}|_1 \leq r} \|D^{\mathbf{k}} h\|_\infty \qquad (75)$$

In addition, for any $0 \leq \gamma_1 < \gamma_2 < \infty$, we define:

$$\mathcal{W}_{r,m}^{\gamma_1, \gamma_2} := \{f : [-1, 1]^m \to \mathbb{R} \mid f \text{ is } r\text{-smooth and } \|f\|_r^s \leq \gamma_2 \text{ and } \|f\|_r^{s,*} \geq \gamma_1\} \qquad (76)$$

Specifically, we denote, $\mathcal{W}_{r,m}^{\gamma_1}$ when $\gamma_2 = 1$. We notice that this set is compact, since it is closed and subset to the compact set $\mathcal{W}_{r,m}$ (see [1]).

Next, we would like to produce a lower bound for the $N$-width of $\mathcal{W}_{r,m}^\gamma$. In [6, 15], in order to achieve a lower bound for the $N$-width of $\mathcal{W}_{r,m}$, two steps are taken. First, they prove that for any $K \subset L^\infty([-1, 1]^m)$, we have: $\tilde{d}_N(K) \geq b_N(K)$. Here, $b_N(K) := \sup_{X_{N+1}} \sup\{\rho \mid \rho \cdot U(X_{N+1}) \subset K\}$ is the Bernstein $N$-width of $K$. The supremum is taken over all $N+1$ dimensional linear subspaces $X_{N+1}$ of $L^\infty([-1, 1]^m)$ and $U(X) := \{f \in X \mid \|f\|_\infty \leq 1\}$ stands for the unit ball of $X$. As a second step, they show that the Bernstein $N$-width of $\mathcal{W}_{r,m}$ is larger than $\Omega(N^{-r/m})$.

Unfortunately, in the general case, Bernstein's $N$-width is very limited in its ability to estimate the nonlinear $N$-width. When considering a set $K$ that is not centered around 0, Bernstein's $N$-width can be arbitrarily smaller than the actual nonlinear $N$-width of $K$. For example, if all of the members of $K$ are distant from 0, then, the Bernstein's $N$-width of $K$ is zero but the nonlinear $N$-width of $K$ that might be large. Specifically, the Bernstein $N$-width of $\mathcal{W}_{r,m}^\gamma$ is small even though intuitively, this set should have a similar width as the standard Sobolev space (at least for a small enough $\gamma > 0$). Therefore, for the purpose of measuring the width of $\mathcal{W}_{r,m}^\gamma$, we define the extended Bernstein $N$-width of a set $K$,

$$\tilde{b}_N(K) := \sup_{X_{N+1}} \sup\{\rho \mid \exists \beta < \rho \text{ s.t } \rho \cdot U(X_{N+1}) \setminus \beta \cdot U(X_{N+1}) \subset K\} \qquad (77)$$

with the supremum taken over all $N + 1$ dimensional linear subspaces $X_{N+1}$ of $L^\infty([-1, 1]^m)$.

The following lemma extends Lem. 3.1 in [6] and shows that the extended Bernstein $N$-width of a set $K$ is a lower bound of the nonlinear $N$-width of $K$.

**Lemma 17.** *Let $K \subset L^\infty([-1, 1]^m)$. Then, $\tilde{d}_N(K) \geq \tilde{b}_N(K)$.*

*Proof.* The proof is based on the proof of Lem. 3.1 in [6]. For completeness, we re-write the proof with minor modifications. Let $\rho < \tilde{b}_N(K)$ and let $X_{N+1}$ be an $N + 1$ dimensional subspace of $L^\infty([-1, 1]^m)$, such that, there exists $0 < \beta < \rho$ and $[\rho \cdot U(X_{N+1}) \setminus \beta \cdot U(X_{N+1})] \subset K$. If $f(\cdot; \theta)$ is class of functions with $N_f = N$ parameters and $S(y)$ is any continuous selection for $K$, such that,

$$\alpha := \sup_{y \in K} \|f(\cdot; S(y)) - y\|_\infty \qquad (78)$$

we let $\hat{S}(y) := S(y) - S(-y)$. We notice that, $\hat{S}(y)$ is an odd continuous mapping of $\partial(\rho \cdot U(X_{N+1}))$ into $\mathbb{R}^N$. Hence, by the Borsuk-Ulam antipodality theorem [4, 13] (see also [7]), there is a function $y_0$ in $\partial(\rho \cdot U(X_{N+1}))$ for which $\hat{S}(y_0) = 0$, i.e. $S(-y_0) = S(y_0)$. We write

$$2y_0 = (y_0 - \ell(\cdot; S(y_0))) - (-y_0 - \ell(\cdot; S(-y_0))) \tag{79}$$

and by the triangle inequality:

$$2\rho = 2\|y_0\|_\infty \leq \|y_0 - \ell(\cdot; S(y_0)\|_\infty + \| - y_0 - \ell(\cdot; S(-y_0)\|_\infty \tag{80}$$

It follows that one of the two functions $y_0, -y_0$ are approximated by $\ell(\cdot; S(y_0))$ with an error $\geq \rho$. Therefore, we have: $\alpha \geq \rho$. Since the lower bound holds uniformly for all continuous selections $S$, we have: $\tilde{d}_N(K) \geq \rho$. $\qquad \square$

**Lemma 18.** *Let $\gamma \in (0,1)$ and $r, m, N \in \mathbb{N}$. We have:*

$$\tilde{d}_N(\mathcal{W}_{r,m}^\gamma) \geq C \cdot N^{-r/m} \tag{81}$$

*for some constant $C > 0$ that depends only on $r$.*

*Proof.* Similar to the proof of Thm. 4.2 in [6] with additional modifications. We fix the integer $r$ and let $\phi$ be a $C^\infty(\mathbb{R}^m)$ function which is one on the cube $[1/4, 3/4]^m$ and vanishes outside of $[-1, 1]^m$. Furthermore, let $C_0$ be such that $1 < \|D^{\mathbf{k}}\phi\|_\infty < C_0$, for all $|\mathbf{k}| < r$. With no loss of generality, we consider integers $N$ of the form $N = d^m$ for some positive integer $d$ and we let $Q_1, \ldots, Q_N$ be the partition of $[-1, 1]^m$ into closed cubes of side length $1/d$. Then, by applying a linear change of variables which takes $Q_j$ to $[-1, 1]^m$, we obtain functions $\phi_1, \ldots, \phi_N$ with $\phi_j$ supported on $Q_j$, such that:

$$\forall \mathbf{k} \text{ s.t } |\mathbf{k}| \leq r : d^{|\mathbf{k}|} \leq \|D^{\mathbf{k}}\phi_j\|_\infty \leq C_0 \cdot d^{|\mathbf{k}|} \tag{82}$$

We consider the linear space $X_N$ of functions $\sum_{j=1}^N c_j \cdot \phi_j$ spanned by the functions $\phi_1, \ldots, \phi_N$. Let $y = \sum_{j=1}^N c_j \cdot \phi_i$. By Lem. 4.1 in [6], for $p = q = \infty$, we have:

$$\|y\|_r^s \leq C_1 \cdot N^{r/m} \cdot \max_{j \in [N]} |c_j| \tag{83}$$

for some constant $C_1 > 0$ depending only on $r$. By definition, for any $x \in Q_j$, we have: $y(x) = c_j \cdot \phi_j(x)$. In particular,

$$\|y\|_\infty = \max_{j \in [N]} \max_{x \in Q_j} |c_j| \cdot \|\phi_j(x)\|_\infty \tag{84}$$

Therefore, by Eq. 82, we have:

$$\max_{j \in [N]} |c_j| \leq \|y\|_\infty \leq C_0 \cdot \max_{j \in [N]} |c_j| \tag{85}$$

Hence,

$$\|y\|_r^s \leq C_1 \cdot N^{r/m} \cdot \|y\|_\infty \tag{86}$$

Then, by taking $\rho := C_1^{-1} \cdot N^{-r/m}$, any $y \in \rho \cdot U(X_N)$ satisfies $\|y\|_r^s \leq 1$. Again, by Lem. 4.1 and Eq. 82, we also have:

$$\|y\|_r^{s,*} \geq C_2 \cdot \|y\|_r^s \geq C_3 \cdot N^{r/m} \cdot \max_{j \in [N]} |c_j| \tag{87}$$

For some constants $C_2, C_3 > 0$ depending only on $r$. By Eq. 85, we obtain:

$$\|y\|_r^{s,*} \geq \frac{\|y\|_\infty \cdot C_3}{C_0} \cdot N^{r/m} \tag{88}$$

Then, for any $\beta > 0$, such that,

$$\gamma < \frac{\beta \cdot C_3}{C_0} \cdot N^{r/m} < 1 \tag{89}$$

we have: $[\rho \cdot U(X_N) \setminus \beta \cdot U(X_N)] \subset \mathcal{W}_{r,m}^\gamma$. Hence, we have:

$$\tilde{d}_N(\mathcal{W}_{r,m}^\gamma) \geq \tilde{b}_N(\mathcal{W}_{r,m}^\gamma) \geq \rho = C_1^{-1} \cdot N^{-r/m} \tag{90}$$

$$\square$$

**Lemma 19.** *Let $\sigma : \mathbb{R} \to \mathbb{R}$ be a piece-wise $C^1(\mathbb{R})$ activation function with $\sigma' \in BV(\mathbb{R})$. Let $f$ be a class of neural networks with $\sigma$ activations. Let $\mathbb{Y} = \mathcal{W}_{r,m}$ and let $\mathcal{W}_{r,m}^{0,\infty} := \{f : [-1,1]^m \to \mathbb{R} \mid f$ is $r$-smooth and $\|f\|_r^s < \infty\}$. Let $\mathcal{F} : \mathcal{W}_{r,m}^{0,\infty} \to \mathcal{W}_{r,m}^{0,\infty}$ be a continuous functional (w.r.t $\| \cdot \|_r^s$). Assume that for any $y \in \mathbb{Y}$ and $\alpha > 0$, if $y + \alpha \cdot \mathcal{F}(y)$ is non-constant, then it cannot be represented as a member of $f$. Then, if $d(f; \mathbb{Y}) \leq \epsilon$, we have:*

$$N_f = \Omega(\epsilon^{-m/r}) \tag{91}$$

*Proof.* Let $\mathbb{Y}_1 = \mathcal{W}_{r,m}^{0.1,1.5} \subset \mathcal{W}_{r,m}^{0,1.5}$ (the selection of $\gamma = 0.1$ is arbitrary). We note that $\mathcal{F}(\mathbb{Y}_1)$ is a compact set as a continuous image of $\mathbb{Y}_1$. Since $\| \cdot \|_r^*$ is a continuous function over $\mathcal{F}(\mathbb{Y}_1)$ (w.r.t norm $\| \cdot \|_r^s$), it attains its maximal value $0 \leq q < \infty$ within $\mathcal{F}(\mathbb{Y}_1)$. By the triangle inequality, for any $y \in \mathbb{Y}_1$, we have:

$$\|y + \epsilon \cdot \mathcal{F}(y)\|_r^* \geq \|y\|_r^* - \epsilon \cdot \|\mathcal{F}(y)\|_r^s \geq 0.1 - \epsilon \cdot q \tag{92}$$

and also,

$$\forall y \in \mathbb{Y}_1 : \|y - y'\|_\infty \leq \epsilon \cdot q \tag{93}$$

We denote $\mathbb{Y}_2 := \{y + \alpha \cdot \mathcal{F}(y) \mid y \in \mathbb{Y}_1\}$. This is a compact set as a continuous image of the function $\mathcal{G}(y) := y + \epsilon \cdot \mathcal{F}(y)$, over the compact set $\mathbb{Y}_1$. In addition, for any constant $\epsilon < 0.1/q$, by Eq. 92, any $y \in \mathbb{Y}_2$ is a non-constant function.

By Eq. 93 and the triangle inequality, we have:

$$\forall y \in \mathbb{Y}_1 : \|f(\cdot; \theta) - y'\|_\infty \leq \|f(\cdot; \theta) - y\|_\infty + \epsilon \cdot q \tag{94}$$

Hence,

$$\sup_{y \in \mathbb{Y}_1} \inf_\theta \|f(\cdot; \theta) - y'\|_\infty \leq \sup_{y \in \mathbb{Y}_1} \inf_\theta \|f(\cdot; \theta) - y\|_\infty + \epsilon \cdot q = d(f; \mathbb{Y}_1) + \epsilon \cdot q \tag{95}$$

In particular,

$$d(f; \mathbb{Y}_2) = \sup_{y' \in \mathbb{Y}_2} \inf_\theta \|f(\cdot; \theta) - y'\|_\infty \leq d(f; \mathbb{Y}_1) + \epsilon \cdot q \tag{96}$$

By the same argument, we can also show that $d(f; \mathbb{Y}_1) \leq d(f; \mathbb{Y}_2) + \epsilon \cdot q$.

By Lem. 16, there is a continuous selector $S : \mathbb{Y}_2 \to \Theta_f$, such that,

$$\sup_{y' \in \mathbb{Y}_2} \|f(\cdot; S(y')) - y'\|_\infty \leq 2 \sup_{y' \in \mathbb{Y}_2} \min_{\theta \in \Theta_f} \|f(\cdot; \theta) - y'\|_\infty + \epsilon \leq 2(d(f; \mathbb{Y}_1) + \epsilon \cdot q) + \epsilon \tag{97}$$

We note that $d(f; \mathbb{Y}_1) \leq 1.5 \cdot d(f; \mathbb{Y}) \leq 1.5\epsilon$. Therefore, we have:

$$\sup_{y' \in \mathbb{Y}_2} \|f(\cdot; S(y')) - y'\|_\infty \leq (4 + 2q)\epsilon \tag{98}$$

In particular, by defining $S(y) = S(y')$ for all $y \in \mathbb{Y}_2$, again by the triangle inequality, we have:

$$\tilde{d}(f; \mathbb{Y}_1) \leq \sup_{y \in \mathbb{Y}_1} \|f(\cdot; S(y)) - y\|_\infty \leq (4 + 2q)\epsilon + \epsilon \leq (5 + 2q)\epsilon \tag{99}$$

By [6], we have:

$$(5 + 2q)\epsilon \geq \tilde{d}(f; \mathbb{Y}_1) \geq \tilde{d}_N(\mathbb{Y}_1) \geq C \cdot N^{-r/m} \tag{100}$$

for some constant $C > 0$ and $N = N_f$. Therefore, we conclude that: $N_f = \Omega(\epsilon^{-m/r})$. $\square$

We note that the definition of $\mathcal{F}(y)$ is very general. In the following theorem we choose $\mathcal{F}(y)$ to be the zero function. An alternative reasonable choice could $\mathcal{F}(y) := \frac{y}{2+y}$.

**Theorem 1.** *Let $\sigma : \mathbb{R} \to \mathbb{R}$ be a piece-wise $C^1(\mathbb{R})$ activation function with $\sigma' \in BV(\mathbb{R})$. Let $f$ be a class of neural networks with $\sigma$ activations. Let $\mathbb{Y} = \mathcal{W}_{r,m}$. Assume that any non-constant $y \in \mathbb{Y}$ is not a member of $f$. Then, if $d(f; \mathbb{Y}) \leq \epsilon$, we have:*

$$N_f = \Omega(\epsilon^{-m/r}) \tag{101}$$

*Proof.* Follows immediate from Lem. 19 with $\mathcal{F}(y)$ being the zero function for all $y \in \mathbb{Y}$.

### 3.4  Proofs of Thms. 2 and 3

**Lemma 20.** *Let $\sigma : \mathbb{R} \to \mathbb{R}$ be universal, piece-wise $C^1(\mathbb{R})$ activation function with $\sigma' \in BV(\mathbb{R})$. Let $\mathcal{E}_{e,q}$ be an neural embedding method. Assume that $\|e\|_1^s \le \ell_1$ for every $e \in e$ and $q$ is a class of $\ell_2$-Lipschitz neural networks with $\sigma$ activations and bounded first layer $\|W_q^1\|_1 \le c$. Let $\mathbb{Y} := \mathcal{W}_{1,m}$. Assume that any non-constant $y \in \mathbb{Y}$ cannot be represented as a neural network with $\sigma$ activations. If the embedding method achieves error $d(\mathcal{E}_{e,q}, \mathbb{Y}) \le \epsilon$, then, the complexity of $q$ is:*

$$N_q = \Omega\left(\epsilon^{-\min(m, 2m_1)}\right) \tag{102}$$

*where the constant depends only on the parameters $c$, $\ell_1$, $\ell_2$, $m_1$ and $m_2$.*

*Proof.* Assume that $N_q = o(\epsilon^{-(m_1+m_2)})$. For every $y \in \mathbb{Y}$, we have:

$$\inf_{\theta_e, \theta_q} \left\| y - q(x, e(I; \theta_e); \theta_q) \right\|_\infty \le \epsilon \tag{103}$$

We denote by $k$ the output dimension of $e$. Let $\sigma \circ W_q^1$ be the first layer of $q$. We consider that $W_q^1 \in \mathbb{R}^{w_1 \times (m_1 + k)}$, where $w_1$ is the size of the first layer of $q$. One can partition the layer into two parts:

$$\sigma(W_q^1(x, e(x; \theta_e))) = \sigma(W_q^{1,1} x + W_q^{1,2} e(I; \theta_e)) \tag{104}$$

where $W_q^{1,1} \in \mathbb{R}^{w_1 \times m_1}$ and $W_q^{1,2} \in \mathbb{R}^{w_1 \times k}$. We divide into two cases.

**Case 1**   Assume that $w_1 = \Omega(\epsilon^{-m_1})$. Then, by the universality of $\sigma$, we can approximate the class of functions $e$ with a class $d$ of neural networks of size $\mathcal{O}(k \cdot \epsilon^{-m_2})$ with $\sigma$ activations. To show it, we can simply take $k$ neural networks of sizes $\mathcal{O}((\epsilon/\ell_1)^{-m_2}) = \mathcal{O}(\epsilon^{-m_2})$ to approximate the $i$'th coordinate of $e$ separately. By the triangle inequality, for all $y \in \mathbb{Y}$, we have:

$$
\begin{aligned}
&\inf_{\theta_d, \theta_q} \left\| y - q(x, d(I; \theta_d); \theta_q) \right\|_\infty \\
&\le \inf_{\theta_e, \theta_d, \theta_q} \left\{ \left\| y - q(x, e(I; \theta_e); \theta_q) \right\|_\infty + \left\| q(x, d(I; \theta_d); \theta_q) - q(x, e(I; \theta_e); \theta_q) \right\|_\infty \right\} \\
&\le \sup_y \inf_{\theta_d} \left\{ \left\| y - q(x, e(I; \theta_e^*); \theta_q^*) \right\|_\infty + \left\| q(x, d(I; \theta_d); \theta_q^*) - q(x, e(I; \theta_e^*); \theta_q^*) \right\|_\infty \right\} \\
&\le \sup_y \inf_{\theta_d} \left\| q(x, d(I; \theta_d); \theta_q^*) - q(x, e(I; \theta_e^*); \theta_q^*) \right\|_\infty + \epsilon
\end{aligned}
\tag{105}
$$

where $\theta_q^*, \theta_e^*$ are the minimizers of $\left\| y - q(x, e(I; \theta_e); \theta_q) \right\|_\infty$. Next, by the Lipschitzness of $q$, we have:

$$\inf_{\theta_d} \left\| q(x, d(I; \theta_d); \theta_q^*) - q(x, e(I; \theta_e^*); \theta_q^*) \right\|_\infty \le \ell_2 \cdot \inf_{\theta_d} \left\| d(I; \theta_d) - e(I; \theta_e^*) \right\|_\infty \le \ell_2 \cdot \epsilon \tag{106}$$

In particular,

$$\inf_{\theta_d, \theta_q} \left\| y - q(x, d(I; \theta_d); \theta_q) \right\|_\infty \le (\ell_2 + 1) \cdot \epsilon \tag{107}$$

By Thm. 1 the size of the architecture $q(x, d(I; \theta_d); \theta_q)$ is $\Omega(\epsilon^{-m})$. Since $N_q = o(\epsilon^{-(m_1+m_2)})$, we must have $k = \Omega(\epsilon^{-m_1})$. Otherwise, the overall size of the neural network $q(x, d(I; \theta_d); \theta_q)$ is $o(\epsilon^{-(m_1+m_2)}) + \mathcal{O}(k \cdot \epsilon^{-m_2}) = o(\epsilon^{-m})$ in contradiction. Therefore, the size of $q$ is at least $w_1 \cdot k = \Omega(\epsilon^{-2m_1})$.

**Case 2**   Assume that $w_1 = o(\epsilon^{-m_1})$. In this case we approximate the class $W_q^{1,2} \cdot e$, where $W_q^{1,2} \in \mathbb{R}^{w_1 \times k}$, where $\|W_q^{1,2}\|_1 \le c$. The approximation is done using a class $d$ of neural networks of size $\mathcal{O}(w_1 \cdot \epsilon^{-m_2})$. By the same analysis of Case 1, we have:

$$\inf_{\theta_d, \theta_q} \left\| y - \tilde{q}(x, d(I; \theta_d); \theta_q) \right\|_\infty \le (\ell_2 + 1) \cdot \epsilon \tag{108}$$

where $\tilde{q} = q'(W_q^{1,1} x + \mathrm{I} \cdot d(I; \theta_d))$ and $q'$ consists of the layers of $q$ excluding the first layer. We notice that $W_q^{1,1} x + \mathrm{I} \cdot d(I; \theta_d)$ can be represented as a matrix multiplication $M \cdot (x, d(I; \theta_d))$, where

$M$ is a block diagonal matrix with blocks $W_q^{1,1}$ and I. Therefore, we achieved a neural network that approximates $y$. However, the overall size of $q(x, d(I; \theta_d); \theta_q)$ is $o(\epsilon^{-(m_1+m_2)}) + \mathcal{O}(w_1 \cdot \epsilon^{-m_2}) = o(\epsilon^{-m})$ in contradiction. $\qquad \square$

**Lemma 21.** *Let $\sigma$ be a universal piece-wise $C^1(\mathbb{R})$ activation function with $\sigma' \in BV(\mathbb{R})$. Let neural embedding method $\mathcal{E}_{e,q}$. Assume that $\|e\|_1^s \leq \ell_1$ and the output dimension of $e$ is $k = \mathcal{O}(1)$ for every $e \in e$. Assume that $q$ is a class of $\ell_2$-Lipschitz neural networks with $\sigma$ activations. Let $\mathbb{Y} := \mathcal{W}_{1,m}$. Assume that any non-constant $y \in \mathbb{Y}$ cannot be represented as neural networks with $\sigma$ activations. If the embedding method achieves error $d(\mathcal{E}_{e,q}, \mathbb{Y}) \leq \epsilon$, then, the complexity of $q$ is:*

$$N_q = \Omega\left(\epsilon^{-m}\right) \tag{109}$$

*where the constant depends only on the parameters $\ell_1$, $\ell_2$, $m_1$ and $m_2$.*

*Proof.* Follows from the analysis in Case 1 of the proof of Lem. 20.

**Theorem 2.** *Let $\sigma : \mathbb{R} \to \mathbb{R}$ be universal, piece-wise $C^1(\mathbb{R})$ activation function with $\sigma' \in BV(\mathbb{R})$. Let $\mathcal{E}_{e,q}$ be an neural embedding method. Assume that $e$ is a class of continuously differentiable neural network $e$ with zero biases and bounded spectral complexity $\mathcal{C}(e) \leq \ell_1$ and $q$ is a class of neural networks $q$ with $\sigma$ activations, bounded spectral complexity $\mathcal{C}(q) \leq \ell_2$ and bounded first layer $\|W_q^1\|_1 \leq c$. Let $\mathbb{Y} := \mathcal{W}_{1,m}$. Assume that any non-constant $y \in \mathbb{Y}$ cannot be represented as a neural network with $\sigma$ activations. If the embedding method achieves error $d(\mathcal{E}_{e,q}, \mathbb{Y}) \leq \epsilon$, then, the complexity of $q$ is:*

$$N_q = \Omega\left(\epsilon^{-\min(m, 2m_1)}\right) \tag{110}$$

*where the constant depends only on the parameters $\ell_1$, $\ell_2$, $m_1$ and $m_2$.*

*Proof.* First, we note that since $\sigma' \in BV(\mathbb{R})$, we have: $\|\sigma'\|_\infty < \infty$. In addition, $\sigma$ is piece-wise $C^1(\mathbb{R})$, and therefore, by combining the two, it is Lipschitz continuous as well. Let $e := e(I; \theta_e)$ and $q := q(x, z; \theta_q)$ be members of $e$ and $q$ respectively. By Lems 3 and 4, we have:

$$\|e\|_\infty = \sup_{I \in \mathcal{I}} \|e(I; \theta_e)\|_1 \leq \ell_1 \cdot \|I\|_1 \leq m_2 \cdot \ell_1 \tag{111}$$

and also

$$\text{Lip}(e) \leq \ell_1 \tag{112}$$

Since the functions $e$ are continuously differentiable, we have:

$$\sum_{1 \leq |\mathbf{k}|_1 \leq 1} \|D^{\mathbf{k}} e\|_\infty \leq \|\nabla e\|_\infty \leq \text{Lip}(e) \leq \ell_1 \tag{113}$$

Hence,

$$\|e\|_1^s \leq (m_2 + 1) \cdot \ell_1 \tag{114}$$

By similar considerations, we have: $\text{Lip}(q) \leq \ell_2$. Therefore, by Lem. 20, we have the desired. $\qquad \square$

**Theorem 3.** *Let $\sigma : \mathbb{R} \to \mathbb{R}$ be a universal, piece-wise $C^1(\mathbb{R})$ activation function with $\sigma' \in BV(\mathbb{R})$ and $\sigma(0) = 0$. Let $\mathcal{E}_{e,q}$ be an neural embedding method. Assume that $e$ is a class of continuously differentiable neural network $e$ with zero biases, output dimension $k = \mathcal{O}(1)$ and bounded spectral complexity $\mathcal{C}(e) \leq \ell_1$ and $q$ is a class of neural networks $q$ with $\sigma$ activations, bounded spectral complexity $\mathcal{C}(q) \leq \ell_2$. Let $\mathbb{Y} := \mathcal{W}_{1,m}$. Assume that any non-constant $y \in \mathbb{Y}$ cannot be represented as a neural network with $\sigma$ activations. If the embedding method achieves error $d(\mathcal{E}_{e,q}, \mathbb{Y}) \leq \epsilon$, then, the complexity of $q$ is:*

$$N_q = \Omega\left(\epsilon^{-(m_1+m_2)}\right) \tag{115}$$

*Proof.* Follows from Lem. 21 and the proof of Thm. 2.

## 3.5 Proof of Thm. 4

**Lemma 22.** *Let $y \in \mathcal{W}_{r,m}$. Then, $\{y_I\}_{I \in \mathcal{I}}$ is compact and $F : I \mapsto y_I$ is a continuous function.*

*Proof.* First, we note that the set $\mathcal{X} \times \mathcal{I} = [-1,1]^{m_1+m_2}$ is compact. Since $y$ is continuous, it is uniformly continuous over $\mathcal{X} \times \mathcal{I}$. Therefore,

$$\lim_{I \to I_0} \|y_I - y_{I_0}\|_\infty = \lim_{I \to I_0} \sup_{x \in \mathcal{X}} \|y(x, I) - y(x, I_0)\|_2 = 0 \tag{116}$$

In particular, the function $F : I \mapsto y_I$ is a continuous function. In addition, since $\mathcal{I} = [-1,1]^{m_2}$ is compact, the image $\{y_I\}_{I \in \mathcal{I}}$ of $F$ is compact as well. $\qquad\square$

**Lemma 23.** *Let $\sigma$ be a universal, piece-wise $C^1(\mathbb{R})$ activation function with $\sigma' \in BV(\mathbb{R})$ and $\sigma(0) = 0$. Let $\hat{\mathbb{Y}} \subset \mathbb{Y} = \mathcal{W}_{r,m}$ be a compact set of functions $y$, such that, $y_I$ cannot be represented as a neural network with $\sigma$ activations, for any $I \in \mathcal{I}$. Then, there are classes $g$ and $f$ of neural networks with $\sigma$ and ReLU activations (resp.), such that, $d(\mathcal{H}_{f,g}; \hat{\mathbb{Y}}) \le \epsilon$ and $N_g = \mathcal{O}\left(\epsilon^{-m_1/r}\right)$, where the constant depends on $m_1, m_2$ and $r$.*

*Proof.* By the universality of $\sigma$, there is a class of neural networks $g$ with $\sigma$ activations of size:

$$N_g = \mathcal{O}\left(\epsilon^{-m_1/r}\right) \tag{117}$$

such that,

$$\forall p \in \mathcal{W}_{r,m_1} : \inf_{\theta_g \in \Theta_g} \|g(\cdot; \theta_g) - p\|_\infty \le \epsilon \tag{118}$$

Let $\mathbb{Y}' := \bigcup_{I \in \mathcal{I}, y \in \hat{\mathbb{Y}}} \{y_I\}$. We note that, $\mathbb{Y}' \subset \mathcal{W}_{r,m_1}$. Therefore,

$$\forall y \in \hat{\mathbb{Y}} \; \forall I \in \mathcal{I} : \inf_{\theta_g \in \Theta_g} \|g(\cdot; \theta_g) - y_I\|_\infty \le \epsilon \tag{119}$$

By Lem. 16, there is a continuous selector $S : \mathbb{Y}' \to \Theta_g$, such that, for any $p \in \mathbb{Y}'$, we have:

$$\|g(\cdot; S(p)) - p\|_\infty \le 2 \inf_{\theta_g \in \Theta_g} \|g(\cdot; \theta_g) - p\|_\infty + \epsilon \le 3\epsilon \tag{120}$$

We notice that the set $\mathcal{I} \times \hat{\mathbb{Y}}$ is compact as a product of two compact sets. Since $y_I$ is continuous with respect to both $(I, y) \in \mathcal{I} \times \hat{\mathbb{Y}}$, we can define a continuous function $S'(I, y) := S(y_I)$. Since $S'$ is continuous over a compact set, it is bounded as well. We denote by $\mathbb{B}$, a closed ball around $0$, in which the image of $S'$ lies. In addition, by the Heine-Cantor theorem, we have:

$$\forall \delta > 0 \; \exists \epsilon > 0 \; \forall I_1, I_2 \in \mathcal{I}, y_1, y_2 \in \hat{\mathbb{Y}} : \tag{121}$$
$$\|(I_1, y_1) - (I_2, y_2)\| \le \delta \implies \|S'(I_1, y_1) - S'(I_2, y_2)\|_2 \le \epsilon$$

where the metric $\|\cdot\|$ is the product metric of $\mathcal{I}$ and $\hat{\mathbb{Y}}$. In particular, we have:

$$\forall \delta > 0 \; \exists \epsilon > 0 \; \forall I_1, I_2 \in \mathcal{I}, y \in \hat{\mathbb{Y}} : \tag{122}$$
$$\|I_1 - I_2\|_2 \le \delta \implies \|S'(I_1, y) - S'(I_2, y)\|_2 \le \epsilon$$

Therefore, since the functions $S'_y(I) := S'(I, y)$ (for any fixed $y$) are uniformly bounded and share the same rate of uniform continuity, by [9], for any $\hat{\epsilon} > 0$, there is a large enough ReLU neural network $f$, such that,

$$\sup_y \inf_{\theta_f \in \Theta_f} \|S'_y(\cdot) - f(\cdot; \theta_f)\|_\infty \le \hat{\epsilon} \tag{123}$$

Since $g(x; \theta_g)$ is continuous over the compact domain, $\mathcal{X} \times \mathbb{B}$, by the Heine-Cantor theorem, $g$ is uniformly continuous. Hence, for any small enough $\hat{\epsilon} > 0$, we have:

$$\forall y \in \hat{\mathbb{Y}} : \inf_{\theta_f \in \Theta_f} \sup_I \|g(\cdot; f(I; \theta_f)) - g(\cdot; S'_y(I))\|_\infty \le \epsilon \tag{124}$$

In particular, by Eqs. 120 and 124 and the triangle inequality, we have the desired:

$$\forall y \in \hat{\mathbb{Y}} \; \forall I \in \mathcal{I} : \inf_{\theta_f \in \Theta_f} \sup_I \|g(\cdot; f(I; \theta_f)) - y_I\|_\infty \le 4\epsilon \tag{125}$$

$\qquad\square$

**Theorem 4.** *Let $\sigma$ be a universal, piece-wise $C^1(\mathbb{R})$ activation function with $\sigma' \in BV(\mathbb{R})$ and $\sigma(0) = 0$. Let $y \in \mathbb{Y} = \mathcal{W}_{r,m}$ be a target function, such that, $y_I$ cannot be represented as a neural network with $\sigma$ activations for all $I \in \mathcal{I}$. Then, there is a class, $\mathfrak{g}$, of neural networks with $\sigma$ activations and a neural network $f(I; \theta_f)$ with ReLU activations, such that, $h(x, I) = g(x; f(I; \theta_f))$ achieves error $\leq \epsilon$ in approximating $y$ and $N_{\mathfrak{g}} = \mathcal{O}\left(\epsilon^{-m_1/r}\right)$.*

*Proof.* Follows immediately for $\hat{\mathbb{Y}} = \{y\}$.

### 3.6 Proof of Thm. 5

**Theorem 5.** *Let $\sigma : \mathbb{R} \to \mathbb{R}$ be a universal Lipschitz continuous activation function, such that, $\sigma(0) = 0$. Let $\mathfrak{g}$ be a class of neural networks with $\sigma$ activations. Let $y \in \mathbb{Y} := \mathcal{W}_{r,m}$ be a target function. Assume that there is a continuous selector $S \in \mathcal{P}_{r,w,c}$ for the class $\{y_I\}_{I \in \mathcal{I}}$ within $\mathfrak{g}$. Then, there is a hypernetwork $h(x, I) = g(x; f(I; \theta_f))$ that achieves error $\leq \epsilon$ in approximating $y$, such that:*

$$
\begin{aligned}
N_f &= \mathcal{O}(w^{1+m_2/r} \cdot \epsilon^{-m_2/r} + w \cdot N_{\mathfrak{g}}) \\
&= \mathcal{O}(\epsilon^{-m_2/r} + \epsilon^{-m_1/r})
\end{aligned}
\tag{126}
$$

*Proof.* We would like to approximate the function $S$ using a neural network $f$ of the specified complexity. Since $S \in \mathcal{P}_{r,w,c}$, we can represent $S$ in the following manner:

$$
S(I) = M \cdot P(I)
\tag{127}
$$

Here, $P : \mathbb{R}^{m_2} \to \mathbb{R}^w$ and $M \in \mathbb{R}^{N_{\mathfrak{g}} \times w}$ is some matrix of bounded norm $\|M\|_1 \leq c$. We recall that any constituent function $P_i$ are in $\mathcal{W}_{r,m_2}$. By [14], such functions can be approximated by neural networks of sizes $\mathcal{O}(\epsilon^{-m_2/r})$ up to accuracy $\epsilon > 0$. Hence, we can approximate $S(I)$ using a neural network $f(I) := M \cdot H(I)$, where $H : \mathbb{R}^{m_2} \to \mathbb{R}^w$, such that, each coordinate $H_i$ is of size $\mathcal{O}(\epsilon^{-m_2/r})$. The error of $f$ in approximating $S$ is therefore upper bounded as follows:

$$
\begin{aligned}
\|M \cdot H(I) - M \cdot P(I)\|_1 &\leq \|M\|_1 \cdot \|H(I) - P(I)\|_1 \\
&\leq c \cdot \sum_{i=1}^{w} |H_i(I) - P_i(I)| \\
&\leq c \cdot w \cdot \epsilon
\end{aligned}
\tag{128}
$$

In addition,

$$
\|M \cdot P(I)\|_1 \leq \|M\|_1 \cdot \|P(I)\|_1 \leq c \cdot w
\tag{129}
$$

Therefore, each one of the output matrices and biases in $S(I)$ is of norm bounded by $c \cdot w$.

Next, we denote by $W^i$ and $b^i$ the weight matrices and biases in $S(I)$ and by $V^i$ and $d^i$ the weight matrices and biases in $f(I)$. We would like to prove by induction that for any $x \in \mathcal{X}$ and $I \in \mathcal{I}$, the activations of $g(x; S(I))$ and $g(x; f(I))$ are at most $\mathcal{O}(\epsilon)$ distant from each other and the norm of these activations is $\mathcal{O}(1)$.

**Base case:** Let $x \in \mathcal{X}$. Since $\mathcal{X} = [-1, 1]^{m_1}$, we have, $\|x\|_1 \leq m_1 =: \alpha^1$. In addition, we have:

$$
\begin{aligned}
\|\sigma(W^1 x + b^1) - \sigma(V^1 x + d^1)\|_1 &\leq L\|(W^1 x + b^1) - (V^1 x + d^1)\|_1 \\
&\leq L\|W^1 - V^1\|_1 \|x\|_1 + \|b^1 - d^1\|_1 \\
&\leq m_1 \cdot L \cdot c \cdot w \cdot \epsilon + c \cdot w \cdot \epsilon \\
&=: \beta^1 \cdot \epsilon
\end{aligned}
\tag{130}
$$

Here, $L$ is the Lipschitz constant of $\sigma$.

**Induction step:** let $x_1$ and $x_2$ be the activations of $g(x; S(I))$ and $g(x; f(I))$ in the $i$'th layer. Assume that there are constants $\alpha^i, \beta^i > 0$ (independent of the size of $g$, $x_1$ and $x_2$), such that,

$\|x_1 - x_2\|_1 \leq \beta^i \cdot \epsilon$ and $\|x_1\|_1 \leq \alpha^i$. Then, we have:

$$
\begin{aligned}
\|\sigma(W^{i+1}x_1 + b^{i+1})\|_1 &= \|\sigma(W^{i+1}x_1 + b^{i+1}) - \sigma(0)\|_1 \\
&\leq L \cdot \|W^{i+1}x_1 + b^{i+1} - 0\|_1 \\
&\leq L \cdot \|W^{i+1}x_1\|_1 + L \cdot \|b^{i+1}\|_1 \\
&\leq L \cdot \|W^{i+1}\|_1 \cdot \|x_1\|_1 + L \cdot c \cdot w \\
&\leq L \cdot c \cdot w(1 + \alpha^i) =: \alpha^{i+1}
\end{aligned}
\tag{131}
$$

and also:

$$
\begin{aligned}
&\|\sigma(W^{i+1} \cdot x_1 + b^{i+1}) - \sigma(V^{i+1}x_2 + d^{i+1})\|_1 \\
\leq& L \cdot \|(W^{i+1} \cdot x_1 + b^{i+1}) - (V^{i+1}x_2 + d^{i+1})\|_1 \\
\leq& L \cdot \|W^{i+1}x_1 - V^{i+1}x_2\|_1 + L \cdot \|b^{i+1} - d^{i+1}\|_1 \\
\leq& L \cdot \|W^{i+1}x_1 - V^{i+1}x_2\|_1 + L \cdot \epsilon \\
\leq& L \cdot (\|W^{i+1}\|_1 \cdot \|x_1 - x_2\|_1 + \|W^{i+1} - V^{i+1}\|_1 \cdot \|x_2\|_1) + L \cdot \epsilon \\
\leq& L \cdot (c \cdot w \cdot \|x_1 - x_2\|_1 + c \cdot w \cdot \epsilon \cdot \|x_2\|_1) + L \cdot \epsilon \\
\leq& L \cdot (c \cdot w \cdot \|x_1 - x_2\|_1 + c \cdot w \cdot \epsilon \cdot (\|x_1\|_1 + \|x_1 - x_2\|_1)) + L \cdot \epsilon \\
\leq& L \cdot (c \cdot w \cdot \beta^i \cdot \epsilon + c \cdot w \cdot \epsilon \cdot (\alpha^i + \beta^i \cdot \epsilon)) + L \cdot \epsilon \\
\leq& L(c \cdot w \cdot (2\beta^i + \alpha^i) + 1) \cdot \epsilon \\
=:& \beta^{i+1} \cdot \epsilon
\end{aligned}
\tag{132}
$$

If $i + 1$ is the last layer, than the application of $\sigma$ is not present. In this case, $\alpha^{i+1}$ and $\beta^{i+1}$ are the same as in Eqs. 131 and 132 except the multiplication by $L$. Therefore, we conclude that $\|g(\cdot; S(I)) - g(x; f(I))\|_\infty = \mathcal{O}(\epsilon)$.

Since $f$ consists of $w$ hidden functions $H_i$ and a matrix $M$ of size $w \cdot N_g$, the total number of trainable parameters of $f$ is: $N_f = \mathcal{O}(w^{1+m_2/r} \cdot \epsilon^{-m_2/r} + w \cdot N_g)$ as desired. $\qquad\square$