[Reviews · NeurIPS 2020]

Review 1

Summary and Contributions: The authors compare computational complexity of two conditioning models for neural networks: embedding-based and hypernetwork-based. The paper develops a an extension of the optimal nonlinear approximation theory to neural nets and aforementioned conditioning models in particular. In a series of elegant theorems the authors prove advantages of the hypernet approach over embeddings: modularity and reduced complexity as the network size increases.

Strengths: The paper presents estimations of network complexity for the two conditioning methods in a highly technical and compelling elaboration of the optimal nonlinear estimation theory. The conclusions have potentially substantial impact. The paper is well organized and is relatively lucid (given the density of the presented details). It offers a guide for understanding hypernetwork advantages and sheds light on modular meta-learning methods in general, thus. The authors make strides to present their argumentation in a clear manner.

Weaknesses: The choice of the embedding network seems a bit ad hoc and not necessarily optimal: First, embeddings usually refer to mappings using shallow nets (suh as word2vec) as in many applications adding more layers don't improve embedding performance. On the other hand, deeper networks can help learn highly performant *representations*, but often that is achieved using more loss functions , such as triplet loss. Would these observations change the conclusions of the paper? The paper is extremely dense in technical detail, and it would be helpful if authors found a way to popularize their approach and present it in the introduction. To be fair, a number of theorems are preceded by explanatory intuitive explanations, but that is not quite done systematically. Some symbols are used without defining them first. The abstract actually is surprisingly terse and not illuminating. Broader context as in the conclusion section would help the readers to appreciate the significance of this work.

Correctness: The technical approach and methodology appear to be sounds and solid.

Clarity: The paper is well written, but substantial amount of technical details could go to teh supplemental material, and more effort made to provide intuitive clarifications.

Relation to Prior Work: Yes

Reproducibility: Yes

Additional Feedback:


Review 2

Summary and Contributions: The paper theoretically compares and analyses the complexity of embedding-based models and the hypernetworks, when the aim is to model a function that has two different inputs: one that is the input from the dataset and one that is a conditioning signal. In particular, the authors compare the minimal parameter complexity needed to obtain a certain error in each of the two alternatives.

Strengths: The discussed problem is very interesting and gives an interesting insight about the reason for the good results obtained by the hypernetworks in many tasks. Moreover, the proposed analysis deeply explores the theory related to the two considered model types.

Weaknesses: The main weakness of the paper is the presentation. In particular, the first part of the manuscript is not pleasant to read and following the flow of the discussion turns out to be complex. In fact, the problem tacked in the proposed work is not clearly presented until the end of the introduction itself. In the first part of the paper, there are also many references to paper and theorems discussed in the subsequent sections, and that makes it even more complex for the reader to follow the discussion. In this regard, in the manuscript, the [5] has been referred as a fundamental starting point for the proposed analysis, but the discussion about what [5] proposes is very limited (few rows section 1.1), and should be significantly extended. Moreover, in the introduction, some important concepts are reported without giving any reference as proof (e.g. “Modularity suggests that the primary network g whose weights are given by f(I) has the same minimal complexity as required for g_I′ in the worst case.”, “However, in hypernetworks, often the layer before the output one is a bottleneck” ) In Section 1.1 the hypernetwork is defined as a model that “using one RNN to generate the weights of another RNN, called the primary network, which performed the actual task”. In my opinion that is not completely correct, since in [David Ha, Andrew Dai, and Quoc V Le. Hypernetworks], the model is defined for CNN and RNN/LSTM to “relax” the weights sharing constraint, therefore it is not limited to the sequential model. Note also that in the base case the meta-network (the one that generates the weights for the primal network) has in input x_t and h_{t-1} as input, and not a conditioning signal. For what concerns sections 2,3 and 4, the lack of readability is also due to the mathematical notation that is not always clear. In the first part of section 3 the concept of “accuracy” is used without any further explanation. Same problem for the "N-width framework" (discussed in [5]) and BV() and C^1 in theorem 1. Furthermore, the authors should indicate that all the proofs are reported in the supplementary material. In general, taking into account also the quantity of information reported in the supplementary material (and the relevance of this information), in my opinion, this work is more suitable to be published in a journal than in a conference where the number of allowed pages is limited.

Correctness: The theoretical part seems correct, even if some points have to be clarified. For what concerns the Experiments, the reported results (in particular the “synthetic experiments” and “experiments on Real-world Dataset”) is questionable since the 2 models (hyper network and embedding-base network) were tested using similar settings, but it is not clear how the authors have chosen the hyper-parameters of the models. For instance, using a particular learning rate / regularization / dropout rate, etc. can significantly impact the capability of a model. Therefore it is important to report how these hyper-parameters were validated, in order to ensure a fair comparison between two models (using the same hyper-parameters for both models is not a fair way to perform the experiments). Moreover, it is important in order to ascribe the observed phenomena to the hyperparameters considered in the various test (number of layers/embedding dimension). Also in this case it is important to say that some figures (fig. 4) are reported in the supplementary material and not in the paper.

Clarity: No, the paper is complex to read, and some parts turn out to be very confusing.

Relation to Prior Work: The literature review is exhaustive, but some important works that are fundamental to understanding the proposed analysis (in particular [5]) should be discussed more deeply.

Reproducibility: Yes

Additional Feedback: --- Update after author response --- since my review significantly differs from the ones provided by the other reviewers I would like to discuss the various points that the authors correctly highlighted in their response. The first one is the clarity of the paper. I've re-read the manuscript, and honestly, in particular, the first part (section 1), to me is very confusing. My feeling is that the many references to concepts and theorems that are discussed in the subsequent sections make it very complex to follow the flow of the discussion. In this regard (related also to the "Prior work" point) only the lines from 83 to 87 are dedicated to explain the idea proposed in [5] and the extension of this proposed in the paper. To me, it is required to provide a more deep explanation about a base work that is crucial to understand one of the central contributions of the paper (as stated in the introduction). For what concerns the methodology of the experimental results, the authors' response still does not explain how the hyperparameters were chosen. They report an interesting study about the varying single hyper-parameter (the learning rate) that is very useful in terms of analysis. But, in order to do a fair comparison between the models, full validation of each model's hyper-parameters is required (the relationship between the various hyperparameters could significantly influence the results). The authors show the results of tests varying a specific hyper-parameter, but also using a very specific set of values for the others. That introduces a bias (how much the reported the results influenced by the other hyper-parameters? are they are valid in general or only using a specific subset of values for the hyperparameters?). In general, the authors in the response said that they will address many points related to the presentation in the next version of the paper. For this reason, and taking into account the other reviews, I raise my overall score from 5 to 6. But, in my opinion, the paper does not deserve a higher score because: - A Paper that discusses complex topics, like this one, should be presented very clearly, and to me, this is not the case. - A good part of the contribution of this paper is related to the proofs that are in the supplementary material. This paper has 18 pages of supplementary material, and so, "shrinking" it in an 8-pages conference paper is not very reasonable in my humble opinion. - The experimental methodology is not completely fair/correct.


Review 3

Summary and Contributions: The main contribution of the paper is the theoretical analysis that shows that for the overall number of trainable parameters in a hypernetwork is much smaller than the number of trainable parameters of a standard neural network. The theoretical analysis is supported by set of experiments that confirm the theoretical part.

Strengths: The paper refers to interesting topic related with the advantages of hyper nets vs. conditioning. The theoretical justification that that under certain conditions that the target model can be smaller than the model with conditioning factor by orders of magnitude. According to my knowledge, it is the first paper that refers to that aspect of hypernets.

Weaknesses: In theoretical considerations authors assumes approximation of the function y: X x I -> R. Currently, hyper networks are successively applied to generative models, like Flows, where we can observe the mapping I x R^K -> R^K. The question occurs, if the theory scales to multidimensional y representation. Also, some empirical evaluation for such cases would be beneficial. Authors provide validation of the assumption 1 in experimental part, but assumption 2 is is not investigated in experiments. From practical point of view this assumption may not be satisfied for some corner cases. The conclusions of the theoretical considerations are supported by experimental part and are consistent with the intuition about conditioning vs. hyper nets. However, from practical perspective, that is the most important is generalization capability for both of the approaches on unseen cases. It would be beneficial to evaluate if the lower number of parameters required for training corresponds to better generalization of the model for unseen cases.

Correctness: I did not study the proofs in supplement carefully, but I didn't notice any incorrect aspects in theoretical and practical methodology.

Clarity: Paper is clear and easy to follow.

Relation to Prior Work: Authors extend the theory of: Ronald A. DeVore, Ralph Howard, and Charles Micchelli. Optimal nonlinear approximation. 383 Manuscripta Math, 1989. and adapt it to theoretical considerations about optimal capabilities of hyper nets vs. conditional models.

Reproducibility: Yes

Additional Feedback: ############# UPDATE ############# The authors answered all of my concerns in rebuttal. In general, I am satisfied with the provided answers. The example used for the generalization case is a bit unlucky, does not show any difference between hypernets and conditioning, but the authors promised to elaborate more in the revised version. I keep voting for acceptance.


Review 4

Summary and Contributions: The paper aims at explaining the success of hypernetworks. It compares hypernetworks with embedding methods, focusing on the complexity, expressed as the number of trainable parameters. In the adopted theoretical model, hypernetworks have significantly lower complexity, as they manifest a certain degree of modularity.

Strengths: The paper presents theoretical analysis of the problem, and follows with experimental evidence that supports the claim.

Weaknesses: The experimental part is hard to follow and poorly ordered: technical details are mixed with the description of the experiments, and high-level overviews are lacking to begin with. Figure 4 is missing, and Figure 1 is not referenced in the text (are these the same figure?)

Correctness: The theoretical framework is adequate. My theoretical understanding of neural networks is limited and I'm unable to verify the correctness.

Clarity: The paper is written clearly with a coherent narrative. I would suggest to newline and center definitions of crucial symbols, esp. used in the later parts of the paper. In the experimental part, frequently inlined technical details make the reading difficult.

Relation to Prior Work: The prior contributions seem adequate.

Reproducibility: Yes

Additional Feedback:

[Author Response · NeurIPS 2020]



Figure I: Varying the lr.  Figure II: Generalization gap.  Figure III: Validating assmp 2.

**To R1:** **Popularizing the approach** We will add a summary of the results in more accessible language, focusing on the more surprising aspects of hypernetworks and how our framework sheds light on these. **A few undefined symbols** We apologize for that. This will be fixed in the next version of the paper. **Abstract** Thanks for the suggestion. In the next version of the paper, we will broaden the context of the abstract similar to the conclusions section.

**To R2:** **Clarity** We will make an effort to make the reading pleasant and would like to respectfully point out that the other three reviewers believe that the paper is well-written and easy to read. **Prior work** We tried to refer [5] in detail throughout the paper. For instance, in the related work section we explain the tools developed in [5] and how we extend them in our paper (Thm. 1). Additionally, we devoted most of Sec. 3 (L 162-182) to discuss the concepts and results of [5]. In [5] they prove the lower bound in Thm. 1 when assuming the existence of a continuous selector (see L 5-9, L 84-87). In our Thm. 1, we prove the lower bound for the case of neural networks, while proving the existence of a continuous selector. Since the existence of a continuous selector is also a cornerstone in the proofs of Thms. 2-5, the analysis in [5] is insufficient to prove these results. We will further discuss these issues in the next version of the paper. **Unsupported statements in the introduction** The word "suggests" is indeed misleading since the statement on minimal complexity is our definition of modularity, see also L 228-237. We will provide references to support the bottleneck statement, e.g., [21]. **The presentation of [Ha et al. 2016]** was meant to introduce the term "primary network", while mentioning the usage for sequences. We apologize for being inaccurate, we will fix it in the next version of the paper. The definitions we use are given explicitly in L 111-116. **Clarifying defs and symbols** In the next version of the paper, we will clarify the definitions of accuracy and N-width. Unfortunately, we omitted the definition of BV, which is given in L 100-103 in the supplementary. $C^1(\mathbb{R})$ stands for the set of continuously differentiable functions over $\mathbb{R}$. **Hyperparams** We did not apply any sort of regularization/normalization (including dropout) on the two models to minimize the number of hyperparameters and since the baseline method is considered more stable than the hypernetwork, which implies that it may require less regularization. Following the review, we conducted a hyperparameter sensitivity test for the learning rate. We compared the two models in the configuration of Fig. 3(a-b) when fixing the depths of $f$ and $e$ to be 4. The results, shown in Fig. I clearly show that the hypernet outperforms the baseline for every learning rate in which the networks provide non-trivial error rates. **Missing Fig. 4** See comments to R4.

**To R3:** **Multidim case** The theory indeed scales to the multi-dimensional case. In this case, if the output dimension is independent of $\epsilon$, we get the exact same bounds. Our colorization experiment in the supplementary corresponds to a multidimensional target function (with three outputs). **The generalization gaps** for the two models are similar, when varying the depth of $f$ and $e$, same setting as Fig. 3(a-b), see Fig. II. We would explore this further. **Assumption 2** To empirically justify this assumption we trained shallow neural networks on MNIST and Fashion MNIST classification with a varying number of hidden neurons, using the MSE loss and one-hot encoding of the labels. As can be seen in Fig. III, the MSE loss strictly decreases when increasing the number of hidden neurons. This is true for a variety of activation functions. To theoretically justify this assumption, we will prove the following lemma:

**Lemma 1** *Assumption 2 holds for $\mathbb{Y} = C([0,1])$ (i.e., the set of continuous functions $y : [0,1] \to \mathbb{R}$) and 2-layered ReLU networks approximators.*

**To R4:** **Fig. 4** is in the supplementary; the paper's reference to it should have been to Fig. 1. We apologize for this.

[Meta-Review · NeurIPS 2020]

This paper analyzes the theoretical complexity of embedding-based models and hypernetworks, the two types of conditional models for neural networks. The motivation is to be able to understand the results in recent literature that suggests that the overall number of trainable parameters needed for a hypernetwork is significantly lower than traditional neural networks with embedding, while achieving similar or better results. The paper's contributions is to develop a theoretical framework that first extends optimal nonlinear approximation theory to neural nets and conditioning models, and proceed to use this as a foundation to prove advantages of hypernetworks over embeddings in terms of comparing the size of the primary network. They demonstrate that hypernetworks exhibit modularity / reduced complexity (although they admit that modularity is not guaranteed to be achievable through SGD optimization). They go one to show that, under common assumptions, the overall number of trainable parameters in a hypernetwork is much smaller than the number of trainable parameters of a standard neural network. They perform simple experiments (MNIST, CIFAR-10, synthetic toy dataset) to validate and complement their theoretical claims. One of the issues of the paper is the presentation due to its theoretical nature, and NeurIPS 8-page limit, that most of the proofs are in the Appendix, although a reviewer points out that the main paper provides high level details to guide the reader through the details intuitively. In the author's rebuttal, they took the feedback seriously and promised to make the intro section tailored for a more general audience, and also improve clarity of the proofs. I recommend that the authors also use the extra page allocated in the camera ready version to move important details back into the main paper and spend some time in optimizing the presentation of this work, as all the reviewers and myself expect the work to have a high impact and would want to make sure that the effort will be made in the presentation of the paper and also presentation of the work, if this were a spotlight or oral talk. The reviewers also suggested other improvements that the authors received and acknowledged, so I expect them to be in the camera ready version as well. A discussion point is whether this paper would benefit as a journal paper, given the length and rigour. Given the high impact nature of this outstanding work, and potential to change and improve the way practitioners and researchers view neural networks through the use of hypernetworks, most reviewers, including myself, would want to see this work accepted at the conference, despite our limitations. I'm inclined to recommend strong acceptance of this work at NeurIPS conference, as I am confident it will make a great addition, and I will leave it with the authors on whether they would continue developing this work in a Journal format afterwards, perhaps after presenting the work and receiving further feedback at NeurIPS.